# Design, Synthesis, In Silico and In Vitro Studies for New Nitric Oxide-Releasing Indomethacin Derivatives with 1,3,4-Oxadiazole-2-thiol Scaffold

**DOI:** 10.3390/ijms22137079

**Published:** 2021-06-30

**Authors:** Alexandru Sava, Frederic Buron, Sylvain Routier, Alina Panainte, Nela Bibire, Sandra Mădălina Constantin, Florentina Geanina Lupașcu, Alin Viorel Focșa, Lenuţa Profire

**Affiliations:** 1Department of Analytical Chemistry, Faculty of Pharmacy, “Grigore T. Popa” University of Medicine and Pharmacy of Iași, 16 University Street, 700115 Iasi, Romania; alexandru.i.sava@umfiasi.ro (A.S.); alina-diana.panainte@umfiasi.ro (A.P.); nela.bibire@umfiasi.ro (N.B.); 2Institut de Chimie Organique et Analytique ICOA, CNRS UMR 7311, Université d’Orléans, 45067 Orléans, France; frederic.buron@univ-orleans.fr; 3Department of Pharmaceutical Chemistry, Faculty of Pharmacy, “Grigore T. Popa” University of Medicine and Pharmacy of Iași, 16 University Street, 700115 Iasi, Romania; constantin.sandra@umfiasi.ro (S.M.C.); florentina-geanina.lupascu@umfiasi.ro (F.G.L.); alin-viorel-v-focsa@d.umfiasi.ro (A.V.F.)

**Keywords:** indomethacin, 1,3,4-oxadiazole-2-thiol, nitric oxide, docking study, cyclooxygenase, inflammation

## Abstract

Starting from indomethacin (IND), one of the most prescribed non-steroidal anti-inflammatory drugs (NSAIDs), new nitric oxide-releasing indomethacin derivatives with 1,3,4-oxadiazole-2-thiol scaffold (NO-IND-OXDs, **8a–p**) have been developed as a safer and more efficient multitarget therapeutic strategy. The successful synthesis of designed compounds (intermediaries and finals) was proved by complete spectroscopic analyses. In order to study the in silico interaction of NO-IND-OXDs with cyclooxygenase isoenzymes, a molecular docking study, using AutoDock 4.2.6 software, was performed. Moreover, their biological characterization, based on in vitro assays, in terms of thermal denaturation of serum proteins, antioxidant effects and the NO releasing capacity, was also performed. Based on docking results, **8k, 8l** and **8m** proved to be the best interaction for the COX-2 (cyclooxygense-2) target site, with an improved docking score compared with celecoxib. Referring to the thermal denaturation of serum proteins and antioxidant effects, all the tested compounds were more active than IND and aspirin, used as references. In addition, the compounds **8c**, **8h**, **8i**, **8m**, **8n** and **8o** showed increased capacity to release NO, which means they are safer in terms of gastrointestinal side effects.

## 1. Introduction

The non-steroidal anti-inflammatory drugs (NSAIDs) are the most prescribed drugs for management of different pathological conditions where inflammation is involved, based on their analgesic, anti-inflammatory and antipyretic effects [1,2].

The inflammation is a defense reaction of the human body to various harmful agents in restoring the body homeostasis [3,4,5,6]. When inflammation persists for a long time, holding the body in a constant state of alert, it may become chronic, with a negative impact on tissue and organs [7]. The clinical consequences of chronic inflammation-driven damage can be severe and include increased risk of many chronic diseases such as: rheumatoid arthritis [8], inflammatory bowel disease [9], metabolic disorders (diabetes mellitus and obesity) [10,11,12], cardiovascular disorders (ischemic heart disease, atherosclerosis) [13,14], neurodegenerative diseases (Parkinson’s and Alzheimer’s disease) [15,16,17] and cancer [18], many of these conditions being life-threatening [19,20]. The current use of NSAIDs in the European Union is associated with a 19% increased risk of hospital admissions for heart failure compared to periods before 1999–2012, as is well documented [21,22,23]. Despite the clinical benefits, the chronic use of NSAIDs is also associated with increased risk of side effects, including gastrointestinal toxicity, renal injury, hepatotoxicity, hypertension, as well as allergic skin reactions [24,25,26,27]. The main mechanism of NSAIDs’ action, which is responsible for both therapeutic and side effects, is inhibition of cyclooxygenase (COX) enzymes in a wide variety of systems, ranging from microsomal enzyme synthesis to different cells and tissues [24,28,29]. Moreover, in the last two decades, based on the beneficial effects of endogenous nitric oxide (NO), NSAIDs with NO-releasing moiety were developed as a new therapeutic strategy for a variety of clinical conditions which involve acute and chronic inflammation [30,31,32,33,34]. Nitric oxide (NO), an endogenous short-lived free radical, is produced in mammalian cells through nitric oxide synthase-mediated conversion of L-arginine to L-citroline [35,36,37,38]. This important signaling molecule has a key role in a wide variety of biological processes such as immune defenses, inflammation, neurotransmission, vasodilatation, platelet adhesion, thrombosis and wound healing [39,40,41]. In addition, NO is intimately involved in regulating all aspects of our lives from waking, digestion, sexual function, perception of pain and pleasure, memory recall and sleeping [40,41]. Moreover, NO is known to have a protective effect on the gastrointestinal tract (GT), based on its properties in stimulating gastric mucus secretion to increase the mucosal blood flow and to inhibit the leukocyte adherence to the vascular endothelium [39,42,43].

The design of hybrid molecules, able to reduce the inflammation by both COX inhibition and NO release could be of great interest. The aim of our study was to develop new nitric oxide-releasing indomethacin derivatives with 1,3,4-oxadiazole-2-thiol scaffold (NO-IND-OXDs), as a safer and more efficient multitarget therapeutic strategy. Among the library of COX inhibitors, indomethacin (IND) retains our attention as it is one of the most prescribed NSAIDs in the management of osteoarthritis, rheumatoid arthritis, episodes of acute gout, ankylosing spondylitis and acute musculoskeletal pain [44,45]. The long-term use of IND could increase the risk for a wide range of side effects such as gastrointestinal irritation, bleeding and ulceration, dizziness, peripheral edema, arterial hypertension, tachycardia, kidney and liver dysfunction, allergic and anaphylactic reactions, increased anxiety, headache [44,46].

On the other hand, the oxadiazole scaffold serves as core for many synthetic compounds of great interest in medicinal chemistry [47,48,49]. Its use offers several advantages: (i) it is an essential part of the pharmacophore, based on its ligand binding role; (ii) it acts as a flat aromatic linker assuring an appropriate molecule orientation; (iii) it induces metabolic stability, water solubility and lower lipophilicity; (iv) it can easily chemically modulate the compounds which contain carbonyl groups such as amides, carbamates, esters and hydoxamic esters [32,50,51]. According to the literature data, the compounds containing the oxadiazole core, generally 1,3,4-oxadiazole motif, have important biological effects such as anti-inflammatory [32,52,53], antioxidant [54,55], antidiabetic [56], anticonvulsant [57], anticancer [58], antitubercular [59,60], antiviral [61], antidepressant [62]. There have already been several approved oxadiazole-based drugs such as: furamizole (strong antibacterial activity, an antibiotic), butalamine (a vasodilator), oxolamine (a cough suppressant), pleconaril (an antiviral), fasiplon (a non-benzodiazepine anxiolitic drug), raltegravir (an antiretroviral drug for the treatment of HIV infection), nesapidil (an anti-arrhythmic drug), zibotentan (an anticancer drug) as well as tiodazosin (an antihypertensive drug) (Figure 1).

In nature, the oxadiazole scaffold has been reported to occur in quisqualic acid (from *Quiqualis fructus*, strong agonist of (AMPA)-subtype glutamate receptor) and phidianidines A and B (from primitive marine organisms, high cytotoxicity against tumor mammalian cell lines) [63,64].

Moreover, the designed compounds contain, between NO precursor and oxadiazole unit, an aromatic linker, substituted with different electron-withdrawing and electron-donating groups. This fragment may serve as a decoy that promotes the binding of the COX to the designed NO-IND-OXDs, according to the reported substrate trap strategies [65,66,67,68].

Based on presented aspects, the chemical modulation of the free carboxyl group of conventional NSAIDs, such as indomethacin, with 1,3,4-oxadiazole scaffold, can provide new drugs with an improved pharmacological profile, in terms of increased efficiency, fewer side effects and reduced ulcerogenic potential. Moreover, the presence of two pharmacophors (indol and 1,3,4-oxadiazole moieties) could increase the ability to inhibit COX enzymes, especially COX-2.

Herein, we present the design and synthesis of some new NO-IND-OXDs, in silico interaction with COX isoenzymes, based on a molecular docking study, as well as the biological characterization using in vitro assays in terms of thermal denaturation of serum proteins, antioxidant effects and the NO-releasing capacity.

## 2. Results and Discussion

### 2.1. Chemistry

Based on reactivity of the carboxyl group of IND, new NO-IND-OXDs **8a–p** were prepared in a few steps (Scheme 1). Reaction of hydroxy-benzaldehydes **1a–o** and 4-(2-hydroxyethyl)phenol **2p** with 1,2-dibromoethane, afforded on an SN_2_ Williamson ether synthesis, gave the corresponding (bromoethoxy)benzaldehyde derivatives **2a–o** and 2-(4-(2-bromoethoxy)phenyl)ethan-1-ol **3p**, respectively. In the next step, the intermediaries **2a–o** were reduced by standard procedure with NaBH_4_ into their corresponding (bromoethoxy)aromatic primary alcohols **3a–o** in a good yield (86–99%). This method offered significant advantages such as mild reaction conditions and easy isolation (simple neutralization and extraction) of the product.

By the metathesis reaction of **3a–p** with silver nitrate in acetonitrile, the corresponding (hydroxyalkyl)phenoxy nitrates **4a–p** were obtained, which were further reacted with thionyl chloride and iodine, respectively, to form the corresponding (halidealkyl)phenoxy nitrates **5a–p**. In turn, the IND derivative **7** was obtained by peptide coupling reaction between IND and hydrazine hydrate when the IND hydrazide **6** was obtained, which further was reacted with carbon disulfide in presence of triethylamine leading to (4-chlorobenzoyl)-5-methoxy-2-methyl-1H-indol-3-yl)methyl)-1,3,4-oxadiazol **7**. Finally, the intermediates **5a–p** were used to alkylate the IND derivative **7** in presence of triethylamine to form the desired NO-IND-OXDs **8a–p**. The common methods for conversion of alcohols to the corresponding alkyl chloride involve activation of hydroxyl group before treatment with the chlorination agent [69]. In the present study we used thionyl chloride in combination with benzotryazole as the activation agent. Benzotryazole is a mild base as well as acid and offers several advantages. It can be removed easily from the reaction mixture by acid or alkali and its hydrochloride salt is insoluble in organic solvents such as dichloromethane [70,71]. Using this method, the desired compounds **5a–m**, **t**, **u** were obtained at room temperature in high yields which ranged between 85 and 99%.

The compound **7** was obtained in near quantitative yield (92%) and was somewhat easier (reaction time of 3 h) while NO-IND-OXDs were obtained at room temperature in good yields ranged between 70 and 91%.

The chemical structure of all synthesized compounds (intermediate and final compounds) was proven on the basis of nuclear magnetic resonance (^1^H NMR, ^13^C NMR) and high-resolution mass spectrometry (HRMS) analysis Appendix A.

In the ^1^H NMR spectra of (halidealkyl)phenoxy nitrates **5a–p** it was found that the multiple signal of protons (-CH_2_ONO_2_) that are in the vicinity of electron-withdrawing nitrate ester group appear more unshielded (δ = 4.7–4.9 ppm, m, 2H), compared with the protons in the vicinity of bromide (-CH_2_Br) (δ = 3.7–3.90 ppm, m, 2H) from (bromoethoxy)aromatic alcohol derivatives **3a–p**. At the same time, ^13^C NMR signal for -CH_2_ONO_2_ was observed at δ = 72–73 ppm, compared to the signal of the same carbon from derivatives **3a–p** (δ = 31–32 ppm). In ^1^H NMR spectra, the final NO-IND-OXDs **8a–p** showed signals for CH_2_ONO_2_ and CH_2_S in the range of δ = 4.7–4.9 ppm, and δ = 4.3–4.5 ppm, respectively. Whereas in ^13^C NMR spectra the signal for CH_2_ONO_2_ group was observed in the range of δ = 71–72 ppm, the signal for CH_2_S was observed in the range of δ = 34–36 ppm and for quaternary carbon from second position of 1,3,4-oxadiazol scaffold (C_q_S) in the range of δ = 162–163 ppm. The proton and carbon signals of other aliphatic and aromatic fragments of **8a–p** were observed at the expected values of chemical shift. The presence of the abovementioned signals have confirmed the presence of 2-mercapto-1,3,4-oxadiazole scaffold and the preserved nitrate ester moiety in the final compounds. Moreover, the NMR spectral data coupled with mass spectra support the proposed structures of the all-synthesized compounds.

### 2.2. In Silico Docking Study

To predict the binding mode of the synthesized compounds with the COX isoenzymes, two X-ray crystallographic structures, obtained from crystallographic data, were used [72]. The PDB 4o1z and PDB 3nt1 were described as the COX-1 and COX-2 structures with the highest resolution to date, 2.4 Å and 1.7 Å, respectively [73], which means that there is more confidence in the location of atoms in the electron density map. Furthermore, the reliability factor was less than 0.16, indicating a strong agreement between the simulated and the experimentally observed diffraction patterns [74,75].

The used computational method was validated using RSMD variation at less than 2 Å for reference drug (indomethacin—IND, diclofenac—DCF and celecoxib—CCB, individually) into the active site of both COX-1 and COX-2 isoforms and the same binding cleft residues reported by other authors were found [76,77,78,79]. The CCB is a preferential COX-2 selective inhibitor whereas IND and DCF are non-selective COX inhibitors, inhibiting both types of the COX enzymes.

Moreover, the docking results showed that all reference drugs have polar interactions with the catalytic site of both COX-1 and COX-2 isoenzymes. More specifically, they act as hydrogen bond acceptors with Arg120 and Tyr355, and have long-range polar interactions with Tyr385 and Ser530 and non-polar interactions with Val349, Trp387, Phe-518, Ile/Val523 and Ala527 [80,81].

For estimating the docking score of NO-IND-OXDs **8a–p** each ligand–receptor complex was subjected to careful analysis for the ideal docked positions, based on the least binding energy scores and maximum number of cluster conformations.

In order to establish the statistical significance of the difference between the ligands while removing the receptor variances from the overall error variance term, a two-way ANOVA test was applied. The results established a statistically significant difference between ligands (*F*(0.5, 18, 37) = 3.2648, *p* = 0.0079, Fcrit = 2.2171). The value of the estimated free binding energy of the compounds **8a–p** was less than −10 kcal/mol and less than that of the reference drugs (IND, DCF and CCB) (Table 1). Therefore, less energy is needed to stabilize the synthesized compounds **8a–p** at the ligand-binding center of the COX receptor.

Based on inhibitory constant (Ki) for COX-1 and COX-2 [82], a selectivity index (log_10_Ki_COX-1_/Ki_COX-2_) was also calculated for docked NO-IND-OXDs **8a–p** (Table 1 and Figure 2). It was noted that the studied compounds have a better interaction with the COX-2 target site, with improved selective index in reference to IND and DCF, except **8n, 8f** and **8h,** which have a better docking score for COX-1 target site. Even more, it was observed that **8m**, **8l** and **8k** proved to be the best interaction COX-2 target sites in reference with CCB (Figure 2).

The COX binding pocket, for both COX-1 and COX-2, is a hydrophobic channel which comprises four α-helices, thus creating a hydrophobic surface to the core of the catalytic domain. In the upper part of the channel, both isoenzymes have two important amino acids, Ser530 and Tyr385. Ser530 influences the COX stereochemistry in prostaglandin synthesis while Tyr385 is involved in the hydroperoxidase activity. Moreover, towards the bottom of the COX active site there are two polar amino acids, Arg120 and Tyr355, which together form a narrow constriction at the entrance of the channel.

The main difference between the active sites of COX isoenzymes consists in the replacement of Ile (Ile434 and Ile523) in COX-1 by the less bulky amino acid Val (Val434 and Val523) in COX-2. A single methylene group (Ile vs. Val) is enough to open a secondary internal hydrophobic side pocket in COX-2 that enlarges the volume of the active site by approximately 25% and gives access to Arg513 replaced in COX-1 by a His.

Referring to the binding of NO-IND-OXDs **8a–p** to COX-1, differences were noted between compounds. However, it was observed that all compounds interact with amino acids both from the narrow constriction and the inside of COX-1 active site (Figure 3a). The binding mode of NO-IND-OXDs **8a–p** to COX-2 side pocket is quite similar (Figure 3b) for all compounds, as they have affinity for the hydrophobic channel. It was noted that the NO-releasing chain interacts with the site pocket (delimited by Val523, Phe518, Arg513, Ala516, Gln192, Tyr355) and extra space (delimited by Leu384, Leu503, Tyr385, Trp387). The oxadiazole scaffold interacts by hydrogen bonds with Arg120 and Tyr355 from the narrow constriction while the indole structure interacts with amino acids from the entrance of the active site.

The analysis of the results revealed that the COX-1/COX-2 selectivity of **8a–p** is influenced by the position of nitrate ester moiety on aromatic ring and by the number and type of the substituents on aromatic ring. The most proper positions of nitrate ester moiety are *meta* and *para*, **8i** (3-oxy-ethylnitrate) and **8a** (4-oxy-ethylnitrate), respectively, being more active on COX-2 than **8n** (2-oxy-ethylnitrate), which is more COX-1-selective. Referring to the substitution of phenoxy-ethylnitrate moiety, the presence of halogens (F, Cl, Br) or NO_2_ on aromatic ring increases the COX-2 selectivity index, in contrast with only OCH_3_ (**8h**, **8f**), which increases COX-1 selectivity. Moreover, it was observed that the elongation of distance between oxadiazole moiety and aromatic ring, from methylene (**8a**) to ethylene (**8p**), seems to be responsible for better ligand flexibility and selectivity for the COX-2 active site. Based on the obtained results, we can say that the synthesized NO-IND-OXDs have the theoretical premises to be promising anti-inflammatory agents. This is in agreement with the literature which reported that the replacement of the free carboxylic acid group of conventional NSAID with 1,3,4-oxadiazole heterocycle may provide new drugs with increased anti-inflammatory activity, improved efficiency and with fewer side effects, reducing ulerogenic potential [53,83].

### 2.3. In Vitro Radical Scavenging Assay

The most used radical scavenging assay is based on the reduction of the DPPH (2,2-diphenyl-1-picrylhydrazyl radical), in the presence of a proton- or electron-donating agent, when there is a color changing from violet to yellow [79]. In order to evaluate the scavenging effect of NO-IND-OXDs, the concentration needed to decrease the initial DPPH concentration (IC_50_) by 50% was calculated and the results are presented in Table 2. Furthermore, the vitamin C equivalent antioxidant capacity (CEAC) value was also calculated, in order to compare the antioxidant effect of **8a–p** to vitamin C (Figure 4).

Using one-way ANOVA test, a statistically significant difference (*F*(17, 71) = 164.778, *p* < 0.05) between tested compounds (**8a–p**, ASP, IND) was highlighted. Post hoc comparisons using the Tukey HSD test revealed that all tested compounds have significantly improved antioxidant effect in reference to IND (*p*< 0.05). Moreover, it was noted that most of the tested compounds (except **8b**, **8l** and **8o**) are significantly more active than aspirin (ASP) (*p*< 0.05). It can be appreciated that the majority of compounds are from 10.0 to 26.7 times more active than IND (CEAC = 0.0014 ± 0.0002) and from 1.5 to 2.3 times more active than ASP (CEAC = 0.0162 ± 0.0004). Compared to vitamin C, used as a positive control, all tested compounds were less active, the CEAC values being less than 1.

Moreover, it was reported that 1,3,4-oxadiazole scaffold has been found to be a potent antioxidant pharmacophore, and chemical modulation of the conventional drugs with this scaffold resulted in a synergistic effect, enhancing the antioxidant efficiency of the parent drug [47,49,50,83,84]. In our research, the presence of 1,3,4-oxadiazole scaffold significantly enhanced antioxidant activity for the majority of the NO-IND-OXDs.

### 2.4. Thermal Denaturation of Serum Proteins

The Mizushima’s test, which is based on thermal denaturation of proteins, was used to predict the anti-inflammatory effects of the new NO-IND-OXDs **8a–p [85]**. The NSAIDs are strong ligands to enzymatic and non-enzymatic proteins and it has been proven that their hydrophilic part, as an anionic radical, interacts with the polar amino acids of proteins while their lipophilic part is fixed into the hydrophobic site of proteins [86]. As a result of these interactions, various changes in the native structure of the proteins, such as modification of the secondary, tertiary or quaternary structure without breaking the covalent bonds, occur [87,88]. It was reported that the native state of the proteins is stable in hydrophobic solvents, whereas the polar organic solvents such as trifluoroethanol, DMFA or DMSO affect their structure, the degree of injury being concentration-dependent [89,90].

When a compound is incubated with non-enzymatic proteins, such as bovine serum albumin (BSA), an increase in optical density will occur because of protein precipitation [88]. Because the tested compounds **8a–p** are less soluble in water, DMSO (40%) was used as solvent. The results showed that BSA (0.2% in 0.9% NaCl/DMSO = 6/4) remained stable after incubation at 38 °C for 5 h and no turbidity was observed. In similar conditions, the compounds **8a**–**p** cause an intense precipitation of BSA and a cloudy solution was obtained. The results, expressed as denaturation effect (%) at different concentration (25 μM, 50 μM and 100 μM) in reference to positive control (100% denaturation of BSA induced by 0.1 M hydrochloric acid) are presented in Figure 5.

Using two-way ANOVA test, it was determined that there is a statistically significant interaction (*F*(30, 144) = 2060.794, *p* < 0.05) between concentration and structure of compounds referring to the denaturation effect. It was noted that the tested compounds **8a**–**p** were able to increase the albumin denaturation in a concentration-dependent manner; the higher concentration increases the denaturation effect. For example, the compounds **8a**, **8i** and **8n** showed a statistically significant increase of denaturation effect (*p* < 0.05) with concentration increasing, but at the same level of each concentration there was no difference (*p* = 0.150) between **8a** and **8i** compared with **8n**, for which the denaturation effect increased significantly less (*p* < 0.05). At 100 µM, compound **8n** was less active (20.7 ± 0.4%) compared with **8a** (33.7 ± 1.0%) and **8i** (33.0 ± 0.9%). Therefore, in relation to chemical structure, we can also note that the presence of nitrate ester moiety in *para* and *meta* positions was more favorable than *ortho*, **8a** (4-oxy-ethylnitrate) and **8i** (3-oxy-ethylnitrate) being more active than **8n** (2-oxy-ethylnitrate).

Compared to the reference drugs, IND and ASP, which have a denaturation effect of less than 5%, all the tested compounds were shown to be more active, with the value of the denaturation effect ranging between 20% and 40% (at 100 µM).

The results of our study also revealed that the presence of halogens (F, Cl, Br) or NO_2_ on phenoxy-ethylnitrate moiety, as well as the elongation of distance between oxadiazole moiety and aromatic ring from methylene group (**8a**) to ethylene group (**8p**) increases the denaturation effect and thus promotes interaction with non-enzymatic proteins.

In addition, it is known that ASP is able to produce acetylation of proteins (such as human serum albumin, fibrinogen, p53, cellular protein) and the ASP therapy is associated with the rise of the anti-acetylated serum albumin antibodies [91,92,93]. It was also reported that these interactions could be used to predict the anti-inflammatory properties of the compounds [94]. The results of our study revealed that the chemical modulation of free carboxylic group of IND with oxadiazole moiety enhances the drug–protein interactions that are in agreement with previous in silico results, which showed good interaction with COX enzyme sites.

### 2.5. In Vitro Nitric Oxide Release Measurement

For detection and quantification of NO released by NO-IND-OXDs **8a**–**p**, a modified colorimetric Griess assay was used, based on decomposition of nitrate ester moiety in presence of Hg^2+^ and thiol-based compounds. The study was performed using different NO donors as reference, such as sodium nitropruside (SNP), nitroglycerine (NTG) and *S*-nitroso-*N*-acetyl-penicillamine (SNAP), which belong to different classes. SNP is a metal nitrosyl compound that spontaneously releases NO at physiological pH, NTG is a representative organic nitrate which requires specific thiol and/or enzymatic activation to generate NO and SNAP is an *S*-nitrosothiol which is rapidly decomposed in the presence of metal ions, such as Cu^+^, Fe^2+^, Hg^2+^ and Ag^+^.

In order to appropriately simulate the body conditions, the NO releasing from the NO-IND-OXDs **8a**–**p** was evaluated in intestinal (PBS, PBS-GSH) and gastric (HCl, HCl-GSH) environmental conditions, respectively, and the results are presented in Figure 6 and Figure 7. As expected, using one-way ANOVA analysis, a statistically significant difference (*F*(2,93) = 60.729, *p* < 0.05) between NO donors, used as reference, was found. In addition, Tukey HSD post hoc comparison test indicated that the NO percentage released for SNAP (52.70 ± 16.82) was significantly higher than the SNP (3.97 ± 1.21) and NTG (0.49 ± 0.18) (Figure 7). There was no statistically significant interaction between experimental conditions (PBS, PBS-GSH, HCl, HCl-GSH) and used Griess reagents (NED—SULF/SULF-HgCl_2_), as determined by two-way Anova (*F*(3.88) = 1.029, *p* = 0.384). Moreover, the presence of Hg^2+^ in Griess reagents increased by a statistically significant value the mean of NO released (*p* = 0.002), but changing the experimental conditions did not significantly influence (*p* = 0.669) the quantified amount of NO. These results suggest that the GSH, present in the experimental medium, fixes the free NO, forming a stable *S*-nitrosothiol (GS-NO) and so prevents the oxidation of NO into stable NO_3_^−^, which cannot be detected using Griess reagents. It was also noted that the Hg^2+^produces the cleavage of the *S*-NO bond from SNAP. Therefore, the GSH and Hg^2+^ are appropriate reagents to measure the total amount of released NO. In addition, the experimental pH has no significant influence on NO release from NO donors.

Referring to the tested compounds **8a**–**p**, no statistically significant interaction (*F*(3, 504) = 0.465, *p* = 0.707) between Griess reagents and experimental conditions, applied to study the NO release, was noted. Moreover, it was noted that the adding of HgCl_2_ to SULF solution did not significantly increase the total amount of NO released from nitrate ester moiety (*F*(1, 504) = 0.122, *p* = 0.727). A statistically significant interaction between structure of compounds **8a**–**p** and experimental conditions (*F*(45, 448) = 69.945, *p* < 0.05) was noted. All these results suggest that the presence of Hg^2+^ in Griess reagents did not influence the amount of NO released by NO-IND-OXDs but it was strongly influenced by experimental conditions (pH and GSH presence) and the structure of tested compounds.

By estimated marginal mean of NO released it was shown that all of the tested compounds **8a–p** released more NO compared with NTG (Figure 7). Using one-way ANOVA test it was determined that there is a statistically significant difference (*F*(16, 255) = 33.054, *p* < 0.05) between tested compounds. Post hoc comparisons using the Tukey HSD test revealed that the tested compounds **8h**, **8n**, **8c**, **8i**, **8m** and **8o**, respectively, released significantly more NO than SNP (*p* < 0.05). It was noted that the amount of NO released is influenced by the position of nitrate ester moiety on aromatic ring and by the number and type of the substituents on aromatic ring. The most proper positions of nitrate ester moiety are *meta* and *ortho*, 8i (3-oxy-ethylnitrate) and 8n (2-oxy-ethylnitrate) being more active than 8a (4-oxy-ethylnitrate). In addition, the elongation of distance between oxadiazole moiety and aromatic ring from methylene group (**8a**) to ethylene group (**8p**) increase the percentage of NO released.

It was reported that NO, at low level (less than 300 nM) exerts beneficial effects (e.g., anti-inflammatory, antioxidant, wound healing effects) while at high level (1000–3000 nM) it has pro-inflammatory effects in pathological conditions [40,95]. The NO-IND-OXDs could act as new drugs because of their balanced inhibition of COX enzymes and their ability to release NO at a submicromolar level, which was proven to be beneficial for inflammation conditions.

## 3. Materials and Methods

### 3.1. Chemistry

#### 3.1.1. General Information

All chemicals used in this research were of analytical grade or HPLC *p.a*. quality, certified by the commercially available sources and were used without further purification unless otherwise specified. The anhydrous solvents were distilled or dried in accordance with standard procedures before use: dichloromethane was distilled under P_2_O_5_ and acetonitrile was dried on cartridge by a GT S100 station. The reaction progress was monitored by thin layer chromatography (TLC), using 2 × 5 cm precoated UV fluorescent silicagel aluminum plate (type Merck 60 F254) and UV lamp exposure (254 nm). The synthesized compounds were purified by flash column chromatography on silicagel 60 (0.063−0.200 mm, Merck), using appropriate solvents. The ^1^H NMR (250 MHz or 400 MHz), ^13^C NMR (63 MHz or 101 MHz) and ^19^F NMR (376 MHz) spectra were recorded on Bruker Avance DPX250 (250, 131 MHz) or Bruker Avance II (400 MHz) spectrometer. Chemical shifts (δ H, δ C) and coupling constant values (*J*) were given in *ppm* and *Hz*, respectively. The chemical shifts were referenced to tetramethylsilane (TMS, δ H = 0, δ C = 0) in CDCl_3_ and DMSO-*d_6_*. The standard abbreviation for signals is as follows: singlet (s), doublet (d), doublet of doublets (dd), doublet of doublets of doublets (ddd), triplet (t), triplet of doublets (dt), quartet (q), hexuplet (h), multiplet (m). For detailed peak assignments, 2D NMR analysis: gradient-selected correlation spectroscopy (gCOSY), gradient-selected heteronuclear multiple bond coherence (gHMBC) and gradient-selected heteronuclear multiple quantum coherence (gHMQC) were used. High-resolution mass spectrometry (HRMS) was recorded on a Bruker maXis mass spectrometer within the “Research Federation” platform between ICOA and CBM (FR2708). The samples were ionized by electrospray ion source (ESI), using N_2_ for nebulization (pressure of 0.6 bar) and drying (flow of 7 L/min, temperature of 200 °C). The capillary potential was set at 4500 V and collision cell RF at 1800.0 Vpp. The ion polarities were positive and were analyzed using scanning range between 50–2500 m/z. Melting points (m.p.) were measured on a Kofler heating bench. A microplate reader (Tecan Sunrise Remote Microplate Reader TW/ML-Abbott F039306) and Hanna Model Pack pH 21 digital pHmeter were also used.

#### 3.1.2. General Procedure for the Synthesis of the (bromoethoxy)benzaldehyde Derivatives (**2a–o**)

The intermediates **2a–o** were obtained by reaction of hydroxybenzaldehydes **1a–o** (1.0 Eq.) in acetonitrile (150 mL for 15 mmol scale) with 1,2-dibromoethane (10.0 Eq.) in the presence of potassium carbonate (2.0 Eq.), according to the method reported in the literature [96,97,98], which was reported in our previous paper [99].

#### 3.1.3. General Procedure for the Synthesis of the (bromoethoxy)aromatic Alcohol Derivatives (**3a–p**)

To a solution of appropriate (bromoethoxy)benzaldehydes **2a–o** (1.0 Eq.), in dry THF (30 mL for 10 mmol scale), sodium borohydride (1.1 Eq.) was added at 0°C and the mixture was stirred at room temperature for 12–24 h, according to the method reported in the literature [100,101,102,103]. Then, the excess of sodium borohydride was quenched by addition of 1M HCl and extracted twice with EtOAc (30 mL). The combined organic layers were washed with water and brine, dried over anhydrous magnesium sulfate, filtered and concentrated under reduced pressure. To obtain *2-(4-(2-bromoethoxy)phenyl)ethanol* (**3p**), 4-(2-hydroxyethyl)phenol (**2p**) (1.0 Eq.) was reacted with 1,2-dibromoethane (10.0 Eq.) in CH_3_CN (50 mL for 15 mmol scale) in the presence of potassium carbonate (2.0 Eq.). The resulting mixture was heated to reflux for 20 h. After cooling to room temperature, distilled water (100 mL) was added and the crude product was extracted with diethyl ether (3 × 50 mL).

The combined organic layers were washed with distilled water (50 mL), brine (50 mL) and dried over anhydrous magnesium sulfate. The crude products were purified by flash column chromatography on silicagel with petroleum ether/EtOAc or CH_2_Cl_2_/CH_3_OH to give the pure compounds **3a**–**p**.

*(4-(2-Bromoethoxy)phenyl)methanol (**3a**)*: white solid, yield 95%, m.p. 73–75°C; ^1^H NMR (250 MHz, DMSO-*d*_6_) δ = 3.68–3.80 (m, 2H), 4.18–4.30 (m, 2H), 4.37 (d, *J* = 5.7 Hz, 2H), 5.01 (t, *J* = 5.7 Hz, 1H), 6.81–6.93 (m, 2H), 7.13–7.25 (m, 2H); ^13^C NMR (101 MHz, DMSO-*d*_6_) δ = 31.5 (CH_2_), 62.5 (CH_2_OH), 67.7 (CH_2_), 114.3 (2CH_Ar_), 127.9 (2CH_Ar_), 135.1 (C_q_), 156.8 (C_q_); HRMS (ESI-MS) m/z calcd for C_9_H_11_BrNaO_2_ [M+Na]^+^: 252.9840, found: 252.9835; Rf (CH_2_Cl_2_/CH_3_OH = 9.8/0.2) 0.53, according to [104].

*(4-(2-Bromoethoxy)-3-fluorophenyl)methanol (**3b**)*: pale yellow oil, yield 93%; ^1^H NMR (250 MHz, DMSO-*d*_6_) δ = 3.76–3.84 (m, 2H), 4.32–4.40 (m, 2H), 4.42 (dd, *J* = 5.7 Hz, 0.7 Hz, 2H), 5.19 (t, *J* = 5.7 Hz, 1H), 7.00–7.22 (m, 3H); ^13^C NMR (101 MHz, DMSO-*d*_6_) δ = 31.2 (CH_2_), 61.9 (CH_2_OH), 69.0 (CH_2_), 114.2 (d, *J* = 18.2 Hz, CH_Ar_), 115.3 (d, *J* = 1.9 Hz, CH_Ar_), 122.4 (d, *J* = 3.4 Hz, CH_Ar_), 136.6 (d, *J* = 5.6 Hz, C_q_), 144.3 (d, *J* = 10.9 Hz, C_q_), 151.6 (d, *J* = 243.9 Hz, C_q_F); ^19^F NMR (376 MHz, DMSO-*d*_6_) δ = -134.7 (dd, *J* = 12.5 Hz, 8.6 Hz); HRMS (ESI-MS) m/z calcd for C_9_H_10_BrFNaO_2_ [M+Na]^+^: 270.9746, found: 270.9740; Rf (CH_2_Cl_2_/CH_3_OH = 9.8/0.2) 0.61.

*(4-(2-Bromoethoxy)-3-chlorophenyl)methanol (**3c**)*: pale yellow oil, yield 97%; ^1^H NMR (400 MHz, DMSO-*d*_6_) δ = 3.75–3.83 (m, 2H), 4.35–4.40 (m, 2H), 4.42 (d, *J* = 5.8 Hz, 2H), 5.19 (t, *J* = 5.8 Hz, 1H), 7.11 (d, *J* = 8.4 Hz, 1H), 7.22 (dd, *J* = 8.4 Hz, 2.0 Hz, 1H), 7.37 (d, *J* = 2.1 Hz, 1H); ^13^C NMR (101 MHz, DMSO-*d*_6_) δ = 31.0 (CH_2_), 61.8 (CH_2_OH), 68.8 (CH_2_), 114.1 (CH_Ar_), 121.2 (C_q_), 126.3 (CH_Ar_), 128.1 (CH_Ar_), 136.6 (C_q_), 152.0 (C_q_). HRMS (ESI-MS) m/z calcd for C_9_H_10_BrClNaO_2_ [M+Na]^+^: 286.9450, found: 286.9835; Rf (petroleum ether/EtOAc = 5/5) 0.58.

*(4-(2-Bromoethoxy)-3,5-dichlorophenyl)methanol (**3d**)*: white solid, yield 96%, m.p. 65–67 °C; ^1^H NMR (250 MHz, DMSO-*d*_6_) δ = 3.72–3.86 (m, 2H), 4.18–4.34 (m, 2H), 4.46 (d, *J* = 5.8 Hz, 2H), 5.40 (t, *J* = 5.8 Hz, 1H), 7.40 (s, 2H); ^13^C NMR (101 MHz, DMSO-*d*_6_) δ = 31.2 (CH_2_), 61.3 (CH_2_OH), 72.9 (CH_2_), 126.8 (2CH_Ar_), 128.0 (C_q_), 141.3 (C_q_), 148.5 (C_q_); HRMS (ESI-MS) m/z calcd for C_9_H_9_BrCl_2_NaO_2_ [M+Na]^+^: 320.9061, found: 320.9055; Rf (CH_2_Cl_2_/CH_3_OH = 9.8/0.2) 0.64.

*(4-(2-Bromoethoxy)-3-nitrophenyl)methanol (**3e**)*: brown oil, yield 86%; ^1^H NMR (400 MHz, DMSO-*d*_6_) δ = 3.72–3.84 (m, 2H), 4.41–4.56 (m, 4H), 5.35 (t, *J* = 5.7 Hz, 1H), 7.34 (d, *J* = 8.7 Hz, 1H), 7.57 (dd, *J* = 8.6 Hz, 2.1 Hz, 1H), 7.79 (d, *J* =2.1 Hz, 1H); ^13^C NMR (101 MHz, DMSO-*d*_6_) δ = 30.7 (CH_2_), 61.4 (CH_2_OH), 69.4 (CH_2_), 115.4 (CH_Ar_), 122.6 (CH_Ar_), 132.2 (CH_Ar_), 135.9 (C_q_), 139.4 (C_q_), 149.3 (C_q_); HRMS (ESI-MS) m/z calcd for C_9_H_11_BrNO_4_ [M+H]^+^: 275.9871, found: 275.9865; Rf (petroleum ether/EtOAc = 4/6) 0.56.

*(4-(2-Bromoethoxy)-3-methoxyphenyl)methanol (**3f**)*: white solid, yield 91%; ^1^H NMR (250 MHz, DMSO-*d*_6_) δ = 3.70–3.82 (m, 2H), 3.77 (s, 3H), 4.26 (m, 2H), 4.42 (d, *J* = 5.7 Hz, 2H), 5.08 (t, *J* = 5.7 Hz, 1H), 6.81 (ddt, *J* = 8.1 Hz, 1.9 Hz, 0.72, 1H), 6.92 (d, *J* = 8.1 Hz, 1H), 6.95 (d, *J* = 1.9 Hz, 1H); ^13^C NMR (101 MHz, DMSO-*d*_6_) δ = 31.5 (CH_2_), 55.5 (CH_3_), 62.7 (CH_2_OH), 69.0 (CH_2_), 111.1 (CH_Ar_), 114.2 (CH_Ar_), 118.6 (CH_Ar_), 136.3 (C_q_), 146.0 (C_q_), 149.1 (C_q_);. HRMS (ESI-MS) m/z calcd for C_10_H_13_BrNaO_3_ [M+Na]^+^: 282.9946, found: 282.9940; Rf (petroleum ether/EtOAc = 3/7) 0.53, according to [105,106].

*(4-(2-Bromoethoxy)-3-ethoxyphenyl)methanol (**3g**)*: white solid, yield 93%, m.p. 53–55 °C; ^1^H NMR (250 MHz, DMSO-*d*_6_) δ = 1.32 (t, *J* = 6.9 Hz, 3H), 3.75 (dd, *J* = 6.0 Hz, 5.2 Hz, 2H), 4.03 (q, *J* = 6.9 Hz, 2H), 4.27 (t, *J* = 6.0 Hz, 5.22 Hz, 2H), 4.41 (d, *J* = 5.7 Hz, 0.6 Hz, 2H), 5.07 (t, *J* = 5.7 Hz, 1H), 6.81 (ddt, *J* = 8.0 Hz, 1.8 Hz, 0.6 Hz, 1H), 6.89–6.96 (m, 2H); ^13^C NMR (101 MHz, DMSO-*d*_6_) δ = 14.8 (CH_3_), 31.5 (CH_2_), 62.6 (CH_2_OH), 64.0 (CH_2_), 69.2 (CH_2_), 112.8 (CH_Ar_), 115.1 (CH_Ar_), 118.8 (CH_Ar_), 136.5 (C_q_), 146.4 (C_q_), 148.5 (C_q_); HRMS (ESI-MS) m/z calcd for C_11_H_15_BrNaO_3_ [M+Na]^+^: 297.0102, found: 297.0096; Rf (petroleum ether/EtOAc = 3/7) 0.55, according to [107].

*(4-(2-Bromoethoxy)-3,5-dimethoxyphenyl)methanol (**3h**)*: white solid, yield 96%, m.p. 71–73 °C; ^1^H NMR (400 MHz, DMSO-*d*_6_) δ = 3.64 (td, *J* = 6.2 Hz, 1.3 Hz, 2H), 3.77 (d, *J* = 1.3 Hz, 6H), 4.11 (td, *J* = 6.2 Hz, 1.3 Hz, 2H), 4.44 (d, *J* = 5.6 Hz, 2H), 5.16 (td, *J* = 5.7 Hz, 1.3 Hz, 1H), 6.64 (s, 2H); ^13^C NMR (101 MHz, DMSO-*d*_6_) δ = 31.6 (CH_2_), 55.9 (2CH_3_), 62.9 (CH_2_OH), 72.3 (CH_2_), 103.4 (2CH_Ar_), 134.4 (C_q_), 138.7 (C_q_), 152.6 (2C_q_); HRMS (ESI-MS) m/z calcd for C_11_H_15_BrNaO_4_ [M+H]^+^: 291.0232, found: 291.0228; Rf (petroleum ether/EtOAc = 2/8) 0.50, according to [108].

*(3-(2-Bromoethoxy)phenyl)methanol (**3i**)*: colorless oil, yield 90%; ^1^H NMR (250 MHz, DMSO-*d*_6_) δ = 3.70–3.87 (m, 2H), 4.16–4.34 (m, 2H), 4.48 (d, *J* = 5.8 Hz, 2H), 5.17 (t, *J* = 5.8 Hz, 1H), 6.76–6.86 (m, 1H), 6.86–6.96 (m, 2H), 7.24 (t, *J* = 8.0 Hz, 1H); ^13^C NMR (101 MHz, DMSO-*d*_6_) δ = 31.5 (CH_2_), 62.7 (CH_2_OH), 67.6 (CH_2_), 112.5 (CH_Ar_), 112.8 (CH_Ar_), 119.0 (CH_Ar_), 129.2 (CH_Ar_), 144.4 (C_q_), 157.9 (C_q_); HRMS (ESI-MS) m/z calcd for C_9_H_11_BrNaO_2_ [M+Na]^+^: 252.9840, found: 252.9834; Rf (petroleum ether/EtOAc = 5/5) 0.58, according to [109].

*(3-(2-Bromoethoxy)-4-methoxyphenyl)methanol (**3j**)*: colorless oil, yield 91%; ^1^H NMR (250 MHz, DMSO-*d*_6_) δ = 3.75 (s, 3H), 3.63–3.90 (m, 2H), 4.27 (dd, *J* = 6.1 Hz, 5.1 Hz, 2H), 4.40 (d, *J =* 5.7 Hz, 2H), 5.05 (t, *J* = 5.7 Hz, 1H), 6.84–6.90 (m, 1H), 6.91 (s, 1H), 6.92–6.96 (m, 1H); ^13^C NMR (101 MHz, DMSO-*d*_6_) δ = 31.4 (CH_2_), 55.7 (CH_3_), 62.6 (CH_2_OH), 68.7 (CH_2_), 112.2 (CH_Ar_), 112.9 (CH_Ar_), 119.6 (CH_Ar_), 135.2 (C_q_), 147.1 (C_q_), 148.0 (C_q_); HRMS (ESI-MS) m/z calcd for C_10_H_13_BrNaO_3_ [M+Na]^+^: 282.9946, found: 282.9937; Rf (petroleum ether/EtOAc = 3/7) 0.64.

*(3-(2-Bromoethoxy)-4-nitrophenyl)methanol (**3k**)*: greenish yellow oil, yield 97%; ^1^H NMR (250 MHz, DMSO-*d*_6_) δ = 3.75–3.86 (m, 2H), 4.44–4.53 (m, 2H), 4.57 (d, *J* = 5.7 Hz, 2H), 5.48 (t, *J* = 5.7 Hz, 1H), 7.09 (ddt, *J* = 8.3 Hz, 1.5 Hz, 0.8 Hz, 1H), 7.23–7.34 (m, 1H), 7.86 (d, *J* = 8.3 Hz, 1H); ^13^C NMR (101 MHz, DMSO-*d*_6_) δ = 30.7 (CH_2_), 62.0 (CH_2_OH), 69.2 (CH_2_), 112.6 (CH_Ar_), 118.4 (CH_Ar_), 125.1 (CH_Ar_), 138.0 (C_q_), 150.3 (C_q_), 150.7 (C_q_); HRMS (ESI-MS) m/z calcd for C_9_H_11_BrNO_4_ [M+H]^+^: 275.9871, found: 275.9866; Rf (petroleum ether/EtOAc = 3/7) 0.54.

*(3-(2-Bromoethoxy)-2-chloro-4-methoxyphenyl)methanol (**3l**)*: white solid, yield 99%, m.p. 76–78 °C; ^1^H NMR (400 MHz, DMSO-*d*_6_) δ = 3.73 (t, *J* = 5.8 Hz, 2H), 3.82 (s, 3H), 4.24 (t, *J* = 5.8 Hz, 2H), 4.48 (d, *J* = 5.0 Hz, 2H), 5.24 (t, *J* = 5.6 Hz, 1H), 7.05 (d, *J* = 8.6 Hz, 1H), 7.23 (t, *J* = 8.5 Hz, 0.9 Hz, 1H); ^13^C NMR (101 MHz, DMSO-*d*_6_) δ = 31.5 (CH_2_), 56.1 (CH_3_), 60.2 (CH_2_OH), 72.4 (CH_2_), 111.2 (CH_Ar_), 123.1 (CH_Ar_), 125.2 (C_q_), 132.3 (C_q_), 142.9 (C_q_), 151.9 (C_q_); HRMS (ESI-MS) m/z calcd for C_10_H_12_BrClNaO_3_ [M+Na]^+^: 316.9556, found: 316.9552; Rf (petroleum ether/EtOAc = 3/7) 0.66.

*(2-Bromo-3-(2-bromoethoxy)-4-methoxyphenyl)methanol (**3m**)*: white solid, yield 99%, m.p. 68–70°C; ^1^H NMR (250 MHz, DMSO-*d*_6_) δ = 3.75 (dd, *J* = 6.2 Hz, 5.6 Hz, 2H), 3.82 (s, 3H), 4.23 (dd, *J* = 6.2 Hz, 5.6 Hz, 2H), 4.45 (dd, *J* = 5.6 Hz, 0.9 Hz, 2H), 5.29 (t, *J* = 5.6 Hz, 1H), 7.10 (d, *J* = 8.6 Hz, 1H), 7.24 (dt, *J* = 8.5 Hz, 0.8 Hz, 1H); ^13^C NMR (101 MHz, DMSO-*d*_6_) δ = 31.5 (CH_2_), 56.2 (CH_3_), 62.5 (CH_2_OH), 72.3 (CH_2_), 111.8 (CH_Ar_), 116.4 (C_q_), 123.3 (CH_Ar_), 133.7 (C_q_), 143.9 (C_q_), 151.7 (C_q_); HRMS (ESI-MS) m/z calcd for C_10_H_12_Br_2_NaO_3_ [M+Na]^+^: 360.9051, found: 360.9039; Rf (petroleum ether/EtOAc = 3/7) 0.63.

*(2-(2-Bromoethoxy)phenyl)methanol (**3n**)*: pale yellow oil, yield 88%; ^1^H NMR (250 MHz, DMSO-*d*_6_) δ = 3.72–3.87 (m, 2H), 4.24–4.38 (m, 2H), 4.56 (dd, *J* = 5.5 Hz, 0.8 Hz, 2H), 4.97 (t, *J* = 5.6 Hz, 1H), 6.88–7.02 (m, 2H), 7.20 (dddt, *J* = 8.1 Hz, 7.5 Hz, 1.8 Hz, 0.6 Hz, 1H), 7.40 (dddd, *J* = 7.4 Hz, 1.9 Hz, 1.0 Hz, 0.6 Hz, 1H; ^13^C NMR (101 MHz, DMSO-*d*_6_) δ = 31.8 (CH_2_), 57.7 (CH_2_OH), 67.9 (CH_2_), 111.5 (CH_Ar_), 120.7 (CH_Ar_), 126.9 (CH_Ar_), 127.5 (CH_Ar_), 130.9 (C_q_), 154.5 (C_q_); HRMS (ESI-MS) m/z calcd for C_9_H_11_BrNaO_2_ [M+Na]^+^: 252.9840, found: 252.9834; Rf (petroleum ether/EtOAc = 5/5) 0.49.

*(5-Bromo-2-(2-bromoethoxy)phenyl)methanol (**3o**)*: pale yellow oil, yield 89%; ^1^H NMR (250 MHz, DMSO-*d*_6_) δ = 3.74–3.84 (m, 2H), 4.26–4.36 (m, 2H), 4.53 (dt, *J* = 5.67 Hz, 0.85 Hz, 2H), 5.19 (t, *J* = 5.6 Hz, 1H), 6.93 (d, *J* = 8.6 Hz, 1H), 7.36 (ddt, *J* = 8.6 Hz, 2.6 Hz, 0.7 Hz, 1H), 7.50 (dt, *J* = 2.6 Hz, 0.9 Hz, 1H); ^13^C NMR (101 MHz, DMSO-*d*_6_) δ = 31.5 (CH_2_), 57.3 (CH_2_OH), 68.1 (CH_2_), 112.5 (C_q_), 113.8 (CH_Ar_), 129.1 (CH_Ar_), 129.8 (CH_Ar_), 133.7 (C_q_), 153.6 (C_q_); HRMS (ESI-MS) m/z calcd for C_9_H_10_Br_2_NaO_2_ [M+Na]^+^: 330.8945, found: 330.8953; Rf (petroleum ether/EtOAc = 5/5) 0.47

*2-(4-(2-Bromoethoxy)phenyl)ethanol (**3p**):* white needle crystals, yield82%, m.p. 65–68 °C; ^1^H NMR (250 MHz, DMSO-*d*_6_) δ = 2.65 (t, *J* =7.1 Hz, 2H), 3.55 (td, *J* = 7.1 Hz, 5.2 Hz, 2H), 3.72–3.83 (m, 2H), 4.21–4.33 (m, 2H), 4.58 (t, *J* = 5.2 Hz, 1H), 6.79–6.93 (m, 2H), 7.06–7.23 (m, 2H); ^13^C NMR (101 MHz, DMSO-*d*_6_) δ = 31.5 (CH_2_), 38.1 (CH_2_), 62.4 (CH_2_OH), 67.7 (CH_2_), 114.4 (2CH_Ar_), 129.8 (2CH_Ar_), 132.0 (C_q_), 156.2 (C_q_); HRMS (ESI-MS) m/z calcd for C_10_H_14_BrO_2_ [M+H]^+^: 245.0177, found: 245.0171; Rf (petroleum ether/EtOAc = 3/7) 0.60.

#### 3.1.4. General Procedure for the Synthesis of the (hydroxyalkyl)phenoxy Nitrates (**4a–p**)

To a solution of the appropriate (bromoethoxy)aromatic alcohol derivatives **3a–p** (1.0 Eq.), in acetonitrile (50 mL for 10 mmol scale), silver nitrate (1.5 Eq.) was added. The mixture was stirred under reflux and darkness for approximately 12 h, according to the methods reported in the literature [110,111,112,113,114,115,116,117], which were adapted to our synthesis in terms of the ratio of reagents, solvent, time of reaction, purification method. At the end of the reaction, brine was added to precipitate the excess of silver nitrate. After filtration through celite, the mixture was extracted with diethyl ether (2 × 50 mL). The combined organic layers were washed with distilled water (50 mL) and brine (50 mL), dried over anhydrous magnesium sulfate, filtered and concentrated under reduced pressure. The crude products were purified by flash chromatography (silicagel, petroleum ether/EtOAc) to obtain the pure products **4a**–**p**.

*2-(4-(Hydroxymethyl)phenoxy)ethyl nitrate (**4a**)*: white solid, yield 97%, m.p. 55–57 °C; ^1^H NMR (250 MHz, DMSO-*d*_6_) δ = 4.23–4.34 (m, 2H), 4.42 (d, *J* = 5.6 Hz, 2H), 4.81–4.91 (m, 2H), 5.05 (t, *J* = 5.6 Hz, 1H), 6.79–6.98 (m, 2H), 7.15–7.31 (m, 2H); ^13^C NMR (101 MHz, DMSO-*d*_6_) δ = 62.5 (CH_2_OH), 64.0 (CH_2_), 72.1 (CH_2_), 114.1 (2CH_Ar_), 127.9 (2CH_Ar_), 135.2 (C_q_), 156.7 (C_q_); HRMS (ESI-MS) m/z calcd for C_9_H_11_NNaO_5_ [M+Na]^+^: 236.0535, found: 236.0526; Rf (petroleum ether/EtOAc = 6/4) 0.62.

*2-(2-Fluoro-4-(hydroxymethyl)phenoxy)ethyl nitrate (**4b**)*: greenish yellow oil, yield 95%; ^1^H NMR (400 MHz, DMSO-*d*_6_) δ = 4.33–4.39 (m, 2H), 4.42 (d, *J* = 5.6 Hz, 2H), 4.84–4.94 (m, 2H), 5.19 (t, *J* = 5.7Hz, 1H), 7.03–7.09 (m, 1H), 7.11–7.18 (m, 2H); ^13^C NMR (101 MHz, DMSO-*d*_6_) δ = 61.9 (CH_2_OH), 65.4 (CH_2_), 71.9 (CH_2_), 114.2 (d, *J* = 18.0 Hz, CH_Ar_), 115.1 (d, *J* = 1.8 Hz, CH_Ar_), 122.4 (d, *J* = 3.5 Hz, CH_Ar_), 136.7 (d, *J* = 5.5 Hz, C_q_), 144.2 (d, *J* = 10.7 Hz, C_q_), 151.5 (d, *J* = 243.9 Hz, C_q_F); ^19^F NMR (376 MHz, DMSO-*d*_6_) δ = −134.94 (dd, *J* = 12.1 Hz, 8.3 Hz); HR.MS (ESI-MS) m/z calcd for C_9_H_10_FNNaO_5_ [M+Na]^+^: 254.0441, found: 254.0436; Rf (CH_2_Cl_2_/CH_3_OH = 9.8/0.2) 0.66.

*2-(2-Chloro-4-(hydroxymethyl)phenoxy)ethyl nitrate (**4c**)*: orange oil, yield 97%; ^1^H NMR (400 MHz, DMSO-*d*_6_) δ = 4.33–4.39 (m, 2H), 4.42 (d, *J* = 5.8 Hz, 2H), 4.85–4.95 (m, 2H), 5.20 (t, *J* = 5.8 Hz, 1H), 7.14 (d, *J* = 8.4 Hz, 1H), 7.23 (dd, *J* = 8.4 Hz, 2.0 Hz, 1H), 7.36 (d, *J* = 2.0 Hz, 1H); ^13^C NMR (101 MHz, DMSO-*d*_6_) δ = 61.8 (CH_2_OH), 65.4 (CH_2_), 71.7 (CH_2_), 114.0 (CH_Ar_), 121.2 (C_q_), 126.3 (CH_Ar_), 128.1 (CH_Ar_), 136.7 (C_q_), 151.9 (C_q_);HRMS (ESI-MS) m/z calcd for C_9_H_10_ClNNaO_5_ [M+Na]^+^: 270.0145, found: 270.0140; Rf (CH_2_Cl_2_/CH_3_OH = 9.8/0.2) 0.63.

*2-(2,6-Dichloro-4-(hydroxymethyl)phenoxy)ethyl nitrate (**4d**)*: greenish yellow oil, yield 99%; ^1^H NMR (250 MHz, DMSO-*d*_6_) δ = 4.20–4.34 (m, 2H), 4.46 (dt, *J* = 5.8 Hz, 0.8 Hz, 2H), 4.78–4.94 (m, 2H), 5.40 (t, *J* = 5.8 Hz, 1H), 7.41 (t, *J* = 0.8 Hz 2H); ^13^C NMR (101 MHz, DMSO-*d*_6_) δ = 61.2 (CH_2_OH), 69.4 (CH_2_), 72.5 (CH_2_), 126.8 (2CH_Ar_), 127.9 (2C_q_), 141.4 (C_q_), 148.4 (C_q_); HRMS (ESI-MS) m/z calcd for C_9_H_9_Cl_2_NNaO_5_ [M+Na]^+^: 303.9755, found: 303.9747; Rf (CH_2_Cl_2_/CH_3_OH = 9.8/0.2) 0.53.

*2-(4-(Hydroxymethyl)-2-nitrophenoxy)ethyl nitrate (**4e**)*: brown greenish yellow solid, yield 95%, m.p. 56–58 °C; ^1^H NMR (400 MHz, DMSO-*d*_6_) δ = 4.44–4.53 (m, 4H), 4.86–4.93 (m, 2H), 5.36 (t, *J* = 5.7 Hz, 1H), 7.36 (d, *J* = 8.6 Hz, 1H), 7.59 (dd, *J* = 8.6 Hz, 2.1 Hz, 1H), 7.80 (d, *J* = 2.1 Hz, 1H); ^13^C NMR (101 MHz, DMSO-*d*_6_) δ = 61.4 (CH_2_OH), 66.1 (CH_2_), 71.4 (CH_2_), 115.4 (CH_Ar_), 122.7 (CH_Ar_), 132.3 (CH_Ar_), 136.0 (C_q_), 139.3 (C_q_), 149.3 (C_q_); HRMS (ESI-MS) m/z calcd for C_9_H_10_N_2_NaO_7_ [M+Na]^+^: 281.0386, found: 281.0381; Rf (CH_2_Cl_2_/CH_3_OH = 9.8/0.2) 0.52.

*2-(4-(hydroxymethyl)-2-methoxyphenoxy)ethyl nitrate (**4f**)*: brown solid, yield 99%, m.p. 51–52 °C; ^1^H NMR (250 MHz, DMSO-*d*_6_) δ = 3.75 (s, 3H), 4.17–4.31 (m, 2H), 4.42 (d, *J* = 5.5 Hz, 2H), 4.78–4.91 (m, 2H), 5.09 (t, *J* = 5.6 Hz, 1H), 6.81 (ddt, *J* = 8.0 Hz, 1.9 Hz, 0.7 Hz, 1H), 6.87–7.00 (m, 2H); ^13^C NMR (101 MHz, DMSO-*d*_6_) δ = 55.4 (CH_3_), 62.7 (CH_2_OH), 65.2 (CH_2_), 72.2 (CH_2_), 110.9 (CH_Ar_), 114.0 (CH_Ar_), 118.5 (CH_Ar_), 136.4 (C_q_), 146.0 (C_q_), 149.0 (C_q_); HRMS (ESI-MS) m/z calcd for C_10_H_13_NNaO_6_ [M+Na]^+^: 266.0641, found: 266.0632; Rf (petroleum ether/EtOAc = 6/4) 0.55.

*2-(2-ethoxy-4-(hydroxymethyl)phenoxy)ethyl nitrate (**4g**)*: greenish yellow solid, yield 99%, m.p. 57–59 °C; ^1^H NMR (250 MHz, DMSO-*d*_6_) δ = 1.31 (t, *J* = 6.98 Hz, 3H), 4.01 (q, *J* = 6.96 Hz, 2H), 4.17–4.29 (m, 2H), 4.41 (d, *J* = 5.1 Hz, 2H), 4.76–4.91 (m, 2H), 5.08 (t, *J* = 5.6 Hz, 1H), 6.81 (ddt, *J* = 8.1 Hz, 1.9 Hz, 0.7 Hz, 1H), 6.94 (d, *J* = 8.1 Hz, 1H), 6.94 (d, *J* = 1.9 Hz, 1H); ^13^C NMR (101 MHz, DMSO-*d*_6_) δ = 14.7 (CH_3_), 62.7 (CH_2_OH), 63.8 (CH_2_), 65.7 (CH_2_), 72.2 (CH_2_), 112.4 (CH_Ar_), 114.9 (CH_Ar_), 118.7 (CH_Ar_), 136.6 (C_q_), 146.3 (C_q_), 148.5 (C_q_); HRMS (ESI-MS) m/z calcd for C_11_H_15_NNaO_6_ [M+Na]^+^: 280.0797, found: 280.0794; Rf (petroleum ether/EtOAc = 6/4) 0.48.

*2-(4-(Hydroxymethyl)-2,6-dimethoxyphenoxy)ethyl nitrate (**4h**)*: greenish yellow solid, yield 99%, m.p. 56–58 °C; ^1^H NMR (250 MHz, DMSO-*d*_6_) δ = 3.74 (s, 6H), 4.04–4.15 (m, 2H), 4.43 (dt, *J* = 5.8 Hz, 0.6 Hz, 2H), 4.67–4.78 (m, 2H), 5.17 (t, *J* = 5.8 Hz, 1H), 6.63 (d, *J* = 0.6 Hz, 2H);^13^C NMR (101 MHz, DMSO-*d*_6_) δ = 55.8 (2CH_3_), 62.9 (CH_2_OH), 68.4 (CH_2_), 72.9 (CH_2_), 103.3 (2CH_Ar_), 134.2 (C_q_), 138.9 (C_q_), 152.6 (2C_q_); HRMS (ESI-MS) m/z calcd for C_11_H_16_NO_7_ [M+H]^+^: 274.0927, found: 274.0922; Rf (petroleum ether/EtOAc = 7/3) 0.43.

*2-(3-(Hydroxymethyl)phenoxy)ethyl nitrate (**4i**)*: colorless oil, yield 89%; ^1^H NMR (250 MHz, DMSO-*d*_6_) δ = 4.16–4.34 (m, 2H), 4.48 (d, *J* = 5.8 Hz, 2H), 4.81–4.90 (m, 2H), 5.17 (t, *J* = 5.8 Hz, 1H), 6.76–6.86 (m, 1H), 6.86–6.96 (m, 2H), 7.24 (t, *J* =8.0 Hz, 1H); ^13^C NMR (101 MHz, DMSO-*d*_6_) δ = 62.8 (CH_2_OH), 67.6 (CH_2_), 72.1 (CH_2_),112.5 (CH_Ar_), 112.8 (CH_Ar_), 119.0 (CH_Ar_), 129.2 (CH_Ar_), 144.4 (C_q_), 157.9 (C_q_); HRMS (ESI-MS) m/z calcd for C_9_H_11_NNaO_5_ [M+Na]^+^: 236.0535, found: 236.0529; Rf (CH_2_Cl_2_/CH_3_OH = 9.6/0.4) 0.39.

*2-(5-(Hydroxymethyl)-2-methoxyphenoxy)ethyl nitrate (**4j**)*: yellow solid, yield 80%, m.p. 51–53 °C; ^1^H NMR (250 MHz, DMSO-*d*_6_) δ = 3.74 (s, 3H), 4.21–4.30 (m, 2H), 4.41 (d, *J* = 5.7 Hz, 2H), 4.81–4.90 (m, 2H), 5.06 (t, *J* = 5.7 Hz, 1H), 6.82–6.98 (m, 3H); ^13^C NMR (101 MHz, DMSO-*d*_6_) δ = 55.6 (CH_3_), 62.6 (CH_2_OH), 65.0 (CH_2_), 72.1 (CH_2_), 112.0 (CH_Ar_), 112.8 (CH_Ar_), 119.7 (CH_Ar_), 135.1 (C_q_), 147.0 (C_q_), 148.0 (C_q_); HRMS (ESI-MS) m/z calcd for C_10_H_13_NNaO_6_ [M+Na]^+^: 266.0641, found: 266.0635; Rf (petroleum ether/EtOAc = 7/3) 0.63.

*2-(5-(Hydroxymethyl)-2-nitrophenoxy)ethyl nitrate (**4k**)*: brown oil, yield 94%; ^1^H NMR (250 MHz, DMSO-*d*_6_) δ = 4.43–4.53 (m, 2H), 4.58 (d, *J* = 5.7 Hz, 2H), 4.85–4.96 (m, 2H), 5.49 (t, *J* = 5.7 Hz, 1H), 7.10 (ddt, *J* = 8.3 Hz, 1.5 Hz, 0.7 Hz, 1H), 7.28–7.33 (m, 1H), 7.87 (d, *J* = 8.3 Hz, 1H); ^13^C NMR (101 MHz, DMSO-*d*_6_) δ = 62.0 (CH_2_OH), 65.9 (CH_2_), 71.3 (CH_2_), 112.5 (CH_Ar_), 118.5 (CH_Ar_), 125.1 (CH_Ar_), 137.9 (C_q_), 150.4 (C_q_), 150.7 (C_q_); HRMS (ESI-MS) m/z calcd for C_9_H_10_N_2_NaO_7_ [M+Na]^+^: 281.0386, found: 281.0380; Rf (CH_2_Cl_2_/CH_3_OH = 9.6/0.4) 0.53.

*2-(2-Chloro-3-(hydroxymethyl)-6-methoxyphenoxy)ethyl nitrate (**4l**)*: white solid, yield 91%, m.p. 57–59 °C; ^1^H NMR (400 MHz, DMSO-*d*_6_) δ = 3.80 (s, 3H), 4.15–4.29 (m, 2H), 4.48 (dd, *J* = 5.6 Hz, 0.8 Hz, 2H), 4.75–4.90 (m, 2H), 5.25 (t, *J* = 5.6 Hz, 1H), 7.06 (d, *J* = 8.6 Hz, 1H), 7.24 (dt, *J* = 8.7 Hz, 0.9 Hz, 1H); ^13^C NMR (101 MHz, DMSO-*d*_6_) δ = 56.0 (CH_3_), 60.2 (CH_2_OH), 68.7 (CH_2_), 72.7 (CH_2_), 111.1 (CH_Ar_), 123.2 (CH_Ar_), 125.2 (C_q_), 132.3 (C_q_), 142.8 (C_q_), 151.9 (C_q_); HRMS (ESI-MS) m/z calcd for C_10_H_12_ClNNaO_6_ [M+Na]^+^: 300.0251, found: 300.0246; Rf (CH_2_Cl_2_/CH_3_OH = 9.6/0.4) 0.55.

*2-(2-Bromo-3-(hydroxymethyl)-6-methoxyphenoxy)ethyl nitrate (**4m**)*: white solid, yield 92%, m.p. 58–60 °C; ^1^H NMR (400 MHz, DMSO-*d*_6_) δ = 3.81 (s, 3H), 4.16–4.25 (m, 2H), 4.44 (dd, *J* = 5.6 Hz, 0.8 Hz, 2H), 4.76–4.87 (m, 2H), 5.29 (t, *J* = 5.6 Hz, 1H), 7.10 (d, *J* = 8.5 Hz, 1H), 7.25 (dt, *J* = 8.5 Hz, 0.8 Hz, 1H); ^13^C NMR (101 MHz, DMSO-*d*_6_) δ = 56.0 (CH_3_), 62.5 (CH_2_OH), 68.6 (CH_2_), 72.7 (CH_2_), 111.8 (CH_Ar_), 116.4 (C_q_), 123.4 (CH_Ar_), 133.7 (C_q_), 143.8 (C_q_), 151.7 (C_q_); HRMS (ESI-MS) m/z calcd for C_10_H_12_BrNNaO_6_ [M+Na]^+^: 343.9746, found: 343.9737; Rf (CH_2_Cl_2_/CH_3_OH = 9.6/0.4) 0.56.

*2-(2-(Hydroxymethyl)phenoxy)ethyl nitrate (**4n**)*: orange oil, yield 93%; ^1^H NMR (400 MHz, DMSO-*d*_6_) δ = 4.25–4.31 (m, 2H), 4.47 (d, *J* = 5.6 Hz, 2H), 4.86–4.92 (m, 2H), 4.98 (t, *J* = 5.6 Hz, 1H), 6.97 (t, *J* = 7.4 Hz, 2H), 7.21 (td, *J* = 7.8 Hz, 1.8 Hz, 1H), 7.39 (dd, *J* = 7.1 Hz, 1.2 Hz, 1H); ^13^C NMR (101 MHz, DMSO-*d*_6_) δ = 57.6 (CH_2_OH), 64.4 (CH_2_), 72.0 (CH_2_), 111.4 (CH_Ar_), 120.8 (CH_Ar_), 127.0 (CH_Ar_), 127.5 (CH_Ar_), 130.7 (C_q_), 154.5 (C_q_); HRMS (ESI-MS) m/z calcd for C_9_H_11_NNaO_5_ [M+Na]^+^: 236.0535, found: 236.0530; Rf (CH_2_Cl_2_/CH_3_OH = 9.6/0.4) 0.62.

*2-(4-Bromo-2-(hydroxymethyl)phenoxy)ethyl nitrate (**4o**)*: greenish yellow oil, yield 95%; ^1^H NMR (250 MHz, DMSO-*d*_6_) δ = 4.26–4.34 (m, 2H), 4.45 (dt, *J* = 5.6 Hz, 0.8 Hz, 2H), 4.85–4.92 (m, 2H), 5.19 (t, *J* = 5.6 Hz, 1H), 6.95 (d, *J* = 8.7 Hz, 1H), 7.38 (ddt, *J* = 8.7 Hz, 2.6 Hz, 0.7 Hz, 1H), 7.49 (dt, *J* = 2.6 Hz, 0.9 Hz, 1H); ^13^C NMR (101 MHz, DMSO-*d*_6_) δ = 57.2 (CH_2_OH), 64.8 (CH_2_), 71.8 (CH_2_), 112.6 (C_q_), 113.7 (CH_Ar_), 129.2 (CH_Ar_), 129.9 (CH_Ar_), 133.6 (C_q_), 153.7 (C_q_); HRMS (ESI-MS) m/z calcd for C_9_H_10_BrNNaO_5_ [M+Na]^+^: 313.9640, found: 313.9634; Rf (CH_2_Cl_2_/CH_3_OH = 9.6/0.4) 0.55.

*2-(4-(2-Hydroxyethyl)phenoxy)ethyl nitrate (**4p**)*: white solid, yield 98%, m.p. 53–55 °C; ^1^H NMR (250 MHz, DMSO-*d*_6_) δ = 2.65 (t, *J* = 7.1 Hz, 2H), 3.55 (td, *J* = 7.1 Hz, 5.2 Hz, 2H), 4.21–4.31 (m, 2H), 4.58 (t, *J* = 5.2 Hz, 1H), 4.82–4.89 (m, 2H), 6.82–6.90 (m, 2H), 7.08–7.18 (m, 2H); ^13^C NMR (101 MHz, DMSO-*d*_6_) δ = 38.1 (CH_2_), 62.4 (CH_2_OH), 64.0 (CH_2_), 72.1 (CH_2_), 114.2 (2CH_Ar_), 129.9 (2CH_Ar_), 132.1 (C_q_), 156.1 (C_q_); HRMS (ESI-MS) m/z calcd for C_10_H_13_NNaO_5_ [M+Na]^+^: 250.0691, found: 250.0697; Rf (petroleum ether/EtOAc = 6/4) 0.63.

#### 3.1.5. General Procedure for the Synthesis of the (halidealkyl)phenoxy Nitrates (**5a–p**)

To a stirred solution of the appropriate (hydroxyalkyl)phenoxy nitrates **4a–o** (1.0 Eq.) in dry CH_2_Cl_2_ (35 mL for 7 mmol scale), 10 mL solution of thionyl chloride (1.2 Eq.) and benzotriazole (BTA, 1.2 Eq.) in dry CH_2_Cl_2_was slowly added, into small portions, according to the methods reported in the literature [70,71], which were adapted to our synthesis in terms of the ratio of reagents, solvent, time of reaction, purification method. Before the reaction was complete, benzotriazole hydrochloride started separating out as a solid. Reaction mixture was stirred further for 20–30 min and after that the solid was filtered off. The filtrate was successively washed with distilled water (2 × 50 mL) and brine (50 mL). To obtain 2-(4-(2-iodooethyl)phenoxy)ethyl nitrate **(5p)**, to a solution of (hydroxyethyl)phenoxy)ethyl nitrate (**4p**) (1.0 Eq.) in CH_2_Cl_2_ (30 mL), imidazole (1.3 Eq.), triphenylphosphine (PPh_3_, 1.3 Eq.) and iodine (1.3 Eq.) were sequentially added at 0 °C, according to the method descried in the literature [118,119,120]. The resulting mixture was stirred at room temperature for 6 h and reaction was quenched by addition of 10 mL of saturated aqueous Na_2_S_2_O_3_ solution. The combined organic layers were dried over magnesium sulfate, filtered and concentrated under reduced pressure to obtain a crude product that was purified by flash chromatography (silicagel, petroleum ether/EtOAc) to produce the pure products **5a–p**.

*2-(4-(Chloromethyl)phenoxy)ethyl nitrate (**5a**)*: yellow oil, yield 99%; ^1^H NMR (400 MHz, DMSO-*d*_6_) δ = 4.28–4.36 (m, 2H), 4.73 (s, 2H), 4.85–4.93 (m, 2H), 6.97 (d, *J* = 8.2 Hz, 2H), 7.38 (d, *J* = 8.2 Hz, 2H); ^13^C NMR (101 MHz, DMSO-*d*_6_) δ = 46.1 (CH_2_), 64.0 (CH_2_), 72.0 (CH_2_), 114.6 (2CH_Ar_), 130.3 (C_q_), 130.4 (2CH_Ar_), 157.8 (C_q_); HRMS (ESI-MS) m/z calcd for C_9_H_10_ClNNaO_4_ [M+Na]^+^: 254.0196, found: 254.0185; Rf (petroleum ether/EtOAc = 7/3) 0.69.

*2-(4-(Chloromethyl)-2-fluorophenoxy)ethyl nitrate* (***5b****)*: pale yellow oil, yield 97%; ^1^H NMR (400 MHz, DMSO-*d*_6_) δ = 4.36–4.43 (m, 2H), 4.72 (s, 2H), 4.86–4.93 (m, 2H), 7.16–7.26 (m, 2H), 7.33 (dd, *J* = 12.3 Hz, 1.7 Hz, 1H); ^13^C NMR (101 MHz, DMSO-*d*_6_) δ = 45.3 (CH_2_), 65.3 (CH_2_), 71.7 (CH_2_), 115.2 (d, *J* = 1.9 Hz, CH_Ar_), 116.7 (d, *J* = 18.7 Hz, CH_Ar_), 125.5 (d, *J* = 3.5 Hz, CH_Ar_), 131.3 (d, *J* = 6.6 Hz, C_q_), 145.7 (d, *J* = 10.6 Hz, C_q_), 151.19 (d, *J* = 244.7 Hz, C_q_F); ^19^F NMR (376 MHz, DMSO-*d*_6_) δ = −134.2 (dd, *J* = 12.3 Hz, 7.5 Hz); HRMS (ESI-MS) m/z calcd for C_9_H_9_ClFNNaO_4_ [M+Na]^+^: 272.0102, found: 272.0096; Rf (petroleum ether/EtOAc = 7/3) 0.69.

*2-(2-Chloro-4-(chloromethyl)phenoxy)ethyl nitrate (**5c**)*: white solid, yield 95%, m.p. 52–54 °C; ^1^H NMR (400 MHz, DMSO-*d*_6_) δ = 4.38–4.46 (m, 2H), 4.73 (s, 2H), 4.89–4.95 (m, 2H), 7.19 (d, *J* = 8.5 Hz, 1H), 7.39 (dd, *J* = 8.5 Hz, 2.2 Hz, 1H), 7.54 (d, *J* = 2.1 Hz, 1H); ^13^C NMR (101 MHz, DMSO-*d*_6_) δ = 45.1 (CH_2_), 65.4 (CH_2_), 71.6 (CH_2_), 114.1 (CH_Ar_), 121.3 (C_q_), 129.0 (CH_Ar_), 130.5 (CH_Ar_), 131.7 (C_q_), 153.1 (C_q_); HRMS (ESI-MS) m/z calcd for C_9_H_9_Cl_2_NNaO_4_ [M+Na]^+^: 287.9806, found: 287.9798; Rf (petroleum ether/EtOAc = 7/3) 0.65.

*2-(2,6-Dichloro-4-(chloromethyl)phenoxy)ethyl nitrate (**5d**)*: white solid, yield 85%, m.p. 49–51 °C; ^1^H NMR (400 MHz, DMSO-*d*_6_) δ = 4.27–4.38 (m, 2H), 4.74 (s, 2H), 4.86–4.93 (m, 2H), 7.62 (s, 2H); ^13^C NMR (101 MHz, DMSO-*d*_6_) δ = 43.9 (CH_2_), 69.5 (CH_2_), 72.4 (CH_2_), 128.2 (2C_q_), 129.7 (2CH_Ar_), 136.3 (C_q_), 149.9 (C_q_); HRMS (ESI-MS) m/z calcd for C_9_H_8_Cl_3_NNaO_4_ [M+Na]^+^: 321.9417, found: 321.9406; Rf (petroleum ether/EtOAc = 8/2) 0.70.

*2-(4-(Chloromethyl)-2-nitrophenoxy)ethyl nitrate (**5e**)*: greenish yellow solid, yield 90%, m.p. 56–58 °C; ^1^H NMR (400 MHz, DMSO-*d*_6_) δ = 4.49–4.55 (m, 2H), 4.81 (s, 2H), 4.86–4.94 (m, 2H), 7.42 (d, *J* = 8.7 Hz, 1H), 7.74 (dd, *J* = 8.7 Hz, 2.3 Hz, 1H), 8.00 (d, *J* = 2.3 Hz, 1H); ^13^C NMR (101 MHz, DMSO-*d*_6_) δ = 44.4 (CH_2_), 66.1 (CH_2_), 71.3 (CH_2_), 115.8 (CH_Ar_), 125.3 (CH_Ar_), 131.0 (C_q_), 133.0 (CH_Ar_), 139.2 (C_q_), 150.4 (C_q_); HRMS (ESI-MS) m/z calcd for C_9_H_9_ClN_2_NaO_6_ [M+Na]^+^: 299.0047, found: 299.0044; Rf (CH_2_Cl_2_) 0.70.

*2-(4-(Chloromethyl)-2-methoxyphenoxy)ethyl nitrate (**5f**)*: white solid, yield 85%, m.p. 54–56 °C; ^1^H NMR (250 MHz, DMSO-*d*_6_) δ = 3.77 (s, 3H), 4.20–4.35 (m, 2H), 4.71 (s, 2H), 4.80–4.91 (m, 2H), 6.96–6.99 (m, 2H), 7.07 (s, 1H); ^13^C NMR (101 MHz, DMSO-*d*_6_) δ = 46.5 (CH_2_), 55.6 (CH_3_), 65.0 (CH_2_), 72.0 (CH_2_), 112.9 (CH_Ar_), 113.7 (CH_Ar_), 121.4 (CH_Ar_), 131.0 (C_q_), 147.3 (C_q_), 149.0 (C_q_). HRMS (ESI-MS) m/z calcd for C_10_H_12_ClNNaO_5_ [M+Na]^+^: 284.0302, found: 284.0299; Rf (petroleum ether/EtOAc = 5/5) 0.76.

*2-(4-(Chloromethyl)-2-ethoxyphenoxy)ethyl nitrate (**5g**)*: white solid, yield 84%, m.p. 57–60 °C; ^1^H NMR (250 MHz, DMSO-*d*_6_) δ = 1.32 (t, *J* = 6.9 Hz, 3H), 4.02 (q, *J* = 6.9 Hz, 2H), 4.24–4.33 (m, 2H), 4.70 (s, 2H), 4.81–4.92 (m, 2H), 6.95–6.99 (m, 2H), 7.06 (d, *J* = 1.7 Hz, 0.6 Hz, 1H); ^13^C NMR (101 MHz, DMSO-*d*_6_) δ = 14.6 (CH_3_), 46.4 (CH_2_), 63.9 (CH_2_), 65.4 (CH_2_), 72.0 (CH_2_), 114.4 (CH_Ar_), 114.5 (CH_Ar_), 121.5 (CH_Ar_), 131.2 (C_q_), 147.6 (C_q_), 148.4 (C_q_); HRMS (ESI-MS) m/z calcd for C_11_H_14_ClNNaO_5_ [M+Na]^+^: 298.0458, found: 298.0456; Rf (petroleum ether/EtOAc = 7/3) 0.72.

*2-(4-(Chloromethyl)-2,6-dimethoxyphenoxy)ethyl nitrate (**5h**)*: white solid, yield 86%, m.p. 59–61 °C; ^1^H NMR (400 MHz, DMSO-*d*_6_) δ = 3.76 (s, 6H), 4.09–4.17 (m, 2H), 4.70 (s, 2H), 4.72–4.78 (m, 2H), 6.78 (s, 2H); ^13^C NMR (101 MHz, DMSO-*d*_6_) δ = 46.6 (CH_2_), 55.9 (2CH_3_), 68.5 (CH_2_), 72.9 (CH_2_), 106.1 (2CH_Ar_), 133.6 (C_q_), 135.6 (C_q_), 152.7 (2C_q_); HRMS (ESI-MS) m/z calcd for C_11_H_15_ClNO_6_ [M+H]^+^: 292.0588, found: 292.0580; Rf (petroleum ether/EtOAc = 7/3) 0.61.

*2-(3-(Chloromethyl)phenoxy)ethyl nitrate (**5i**)*: pale yellow oil, yield 88%; ^1^H NMR (250 MHz, DMSO-*d*_6_) δ = 4.26–4.37 (m, 2H), 4.72 (s, 2H), 4.83–4.95 (m, 2H), 6.94 (ddd, *J* = 8.2 Hz, 2.6 Hz, 1.0 Hz, 1H), 7.00–7.09 (m, 2H), 7.24–7.38 (m, 1H); ^13^C NMR (101 MHz, DMSO-*d*_6_) δ = 45.9 (CH_2_), 64.0 (CH_2_), 71.9 (CH_2_), 114.4 (CH_Ar_), 114.9 (CH_Ar_), 121.6 (CH_Ar_), 129.8 (CH_Ar_), 139.2 (C_q_), 157.9 (C_q_); HRMS (ESI-MS) m/z calcd for C_9_H_11_ClNO_4_ [M+H]^+^: 232.0377, found: 232.0371; Rf (petroleum ether/EtOAc = 8/2) 0.60.

*2-(5-(Chloromethyl)-2-methoxyphenoxy)ethyl nitrate (**5j**)*: yellow oil, yield 92%, ^1^H NMR (250 MHz, DMSO-*d*_6_) δ = 3.77 (s, 3H), 4.22–4.33 (m, 2H), 4.70 (s, 2H), 4.79–4.92 (m, 2H), 6.97 (d, *J* = 8.2 Hz, 1H), 7.02 (d, *J* = 1.9 Hz, 1H), 7.09 (d, *J* = 1.9 Hz, 1H); ^13^C NMR (101 MHz, DMSO-*d*_6_) δ = 46.5 (CH_2_), 55.6 (CH_3_), 65.1 (CH_2_), 72.0 (CH_2_), 112.1 (CH_Ar_), 114.7 (CH_Ar_), 122.5 (CH_Ar_), 129.9 (C_q_), 147.1 (C_q_), 149.2 (C_q_); HRMS (ESI-MS) m/z calcd for C_10_H_12_ClNNaO_5_ [M+Na]^+^: 284.0302, found: 284.0297; Rf (petroleum ether/EtOAc = 8/2) 0.45.

*2-(5-(Chloromethyl)-2-nitrophenoxy)ethyl nitrate (**5k**)*: brown solid, yield 91%, m.p. 51–53 °C; ^1^H NMR (400 MHz, DMSO-*d*_6_) δ = 4.47–4.56 (m, 2H), 4.81 (s, 2H), 4.88–4.97 (m, 2H), 7.23 (dd, *J* = 8.3 Hz, 1.6 Hz, 1H), 7.50 (d, *J* = 1.6 Hz, 1H), 7.91 (d, *J* = 8.3 Hz, 1H); ^13^C NMR (101 MHz, DMSO-*d*_6_) δ = 44.7 (CH_2_), 66.1 (CH_2_), 71.3 (CH_2_), 115.6 (CH_Ar_), 121.5 (CH_Ar_), 125.5 (CH_Ar_), 139.0 (C_q_), 144.3 (C_q_), 150.5 (C_q_); HRMS (ESI-MS) m/z calcd for C_9_H_9_ClN_2_NaO_6_ [M+Na]^+^: 299.0047, found: 299.0043; Rf (petroleum ether/CH_2_Cl_2_ = 3/7) 0.67.

*2-(2-Chloro-3-(chloromethyl)-6-methoxyphenoxy)ethyl nitrate (**5l**)*: pale yellow oil, yield 96%; ^1^H NMR (400 MHz, DMSO-*d*_6_) δ = 3.83 (s, 3H), 4.21–4.30 (m, 2H), 4.79 (s, 2H), 4.80–4.87 (m, 2H), 7.08 (d, *J* = 8.6 Hz, 1H), 7.36 (d, *J* = 8.6 Hz, 1H);. ^13^C NMR (101 MHz, DMSO-*d*_6_) δ = 44.2 (CH_2_), 56.2 (CH_3_), 68.8 (CH_2_), 72.7 (CH_2_), 111.4 (CH_Ar_), 126.9 (CH_Ar_), 127.6 (C_q_), 127.7 (C_q_), 143.4 (C_q_), 153.5 (C_q_); HRMS (ESI-MS) m/z calcd for C_10_H_11_Cl_2_NNaO_5_ [M+Na]^+^: 317.9912, found: 317.9906; Rf (petroleum ether/EtOAc = 7/3) 0.67.

*2-(2-Bromo-3-(chloromethyl)-6-methoxyphenoxy)ethyl nitrate (**5m**)*: greenish yellow oil, yield 95%; ^1^H NMR (400 MHz, DMSO-*d*_6_) δ = 3.83 (s, 3H), 4.19–4.30 (m, 2H), 4.80 (s, 2H), 4.81–4.88 (m, 2H), 7.12 (d, *J* = 8.5 Hz, 1H), 7.38 (d, *J* = 8.5 Hz, 1H); ^13^C NMR (101 MHz, DMSO-*d*_6_) δ = 46.8 (CH_2_), 56.2 (CH_3_), 68.7 (CH_2_), 72.7 (CH_2_), 112.1 (CH_Ar_), 119.3 (C_q_), 127.3 (CH_Ar_), 129.3 (C_q_), 144.5 (C_q_), 153.3 (C_q_); HRMS (ESI-MS) m/z calcd for C_10_H_11_BrClNNaO_5_ [M+Na]^+^: 361.9407, found: 361.9401; Rf (petroleum ether/EtOAc = 6/4) 0.71.

*2-(2-(Chloromethyl)phenoxy)ethyl nitrate (**5n**)*: yellow oil, yield 83%; ^1^H NMR (250 MHz, DMSO-*d*_6_) δ = 4.34–4.42 (m, 2H), 4.69 (s, 2H), 4.89–4.97 (m, 2H), 6.92–7.11 (m, 2H), 7.29–7.46 (m, 2H); ^13^C NMR (101 MHz, DMSO-*d*_6_) δ = 40.5 (CH_2_), 64.8 (CH_2_), 72.0 (CH_2_), 111.4 (CH_Ar_), 114.8 (CH_Ar_), 127.5 (CH_Ar_), 129.5 (CH_Ar_), 132.7 (C_q_), 154.5 (C_q_); HRMS (ESI-MS) m/z calcd for C_9_H_10_ClNNaO_4_ [M+Na]^+^: 254.0196, found: 254.0193; Rf (petroleum ether/EtOAc = 8/2) 0.64.

*2-(4-Bromo-2-(bromomethyl)phenoxy)ethyl nitrate (**5o**)*: pale yellow oil, yield 89%; ^1^H NMR (400 MHz, DMSO-*d*_6_) δ = 4.33–4.41 (m, 2H), 4.67 (s, 2H), 4.87–4.96 (m, 2H), 7.07 (d, *J* = 8.7 Hz, 1H), 7.52 (dd, *J* = 8.8 Hz, 2.5 Hz, 1H), 7.62 (d, *J* = 2.6 Hz, 1H); ^13^C NMR (101 MHz, DMSO-*d*_6_) δ = 40.4 (CH_2_), 65.2 (CH_2_), 71.7 (CH_2_), 112.3 (C_q_), 114.9 (CH_Ar_), 128.3 (C_q_), 132.7 (CH_Ar_), 133.1 (CH_Ar_), 155.2 (C_q_); HRMS (ESI-MS) m/z calcd for C_9_H_9_BrClNNaO_4_ [M+Na]^+^: 331.9301, found: 331.9310; Rf (petroleum ether/EtOAc = 8/2) 0.55.

*2-(4-(2-Iodooethyl)phenoxy)ethyl nitrate (**5p**)*: white solid, yield 79%, m.p. 57–59 °C; ^1^H NMR (400 MHz, DMSO-*d*_6_) δ = 3.05 (t, *J* = 7.4 Hz, 2H), 3.42 (t, *J* = 7.4 Hz, 2H), 4.23–4.31 (m, 2H), 4.81–4.91 (m, 2H), 6.89 (d, *J* = 7.4Hz, 2H), 7.18 (d, *J* = 7.6 Hz, 2H); ^13^C NMR (101 MHz, DMSO-*d*_6_) δ = 8.7 (CH_2_), 38.4 (CH_2_), 63.9 (CH_2_), 72.0 (CH_2_), 114.4 (2CH_Ar_), 129.6 (2CH_Ar_), 133.3 (C_q_), 156.5 (C_q_); HRMS (ESI-MS) m/z calcd for C_10_H_13_INO_4_ [M+H]^+^: 337.9889, found: 337.9884; Rf (petroleum ether/EtOAc = 8/2) 0.60.

#### 3.1.6. Synthesis of (4-chlorobenzoyl)-5-methoxy-2-methyl-1H-indol-3-yl)methyl)-1,3,4-oxadiazol (**7**)

To a suspension of indomethacin hydrazide (**6**) [99] (1.0 Eq.) in CH_3_CN (150 mL for 12 mmol scale), Et_3_N (2.0 Eq.) and CS_2_ (2.0 Eq.) were slowly added, into small portions, according to the method reported in the literature [121,122,123,124], which was adapted to our synthesis in terms of the ratio of reagents, solvent, time of reaction and purification method. The mixture was stirred for 3 h under reflux till hydrogen sulfide formation was stopped. After cooling at room temperature, the solvent was removed under reduced pressure and the residue was dissolved in EtOAc (40 mL) and acidified with aqueous diluted HCl 0.5 M solution (10 mL). The organic layer was separated and was successively washed with distilled water (3 × 50 mL) and finally with brine (50 mL). The combined organic layers were dried over anhydrous magnesium sulfate, filtered and concentrated under reduced pressure. The crude product was purified by flash column chromatography to give the pure product: white solid, yield 92%, m.p. 199–201 °C; ^1^H NMR (400 MHz, DMSO-*d*_6_) δ = 2.29 (s, 3H), 3.76 (s, 3H), 4.25 (s, 2H), 6.74 (dd, *J* = 9.0 Hz, 2.5 Hz, 1H), 6.90 (d, *J* = 9.0 Hz, 1H), 7.12 (d, *J* = 2.5 Hz, 1H), 7.56–7.76 (m, 4H), 14.36 (s, 1H); ^13^C NMR (101 MHz, DMSO-*d*_6_) δ = 13.1 (CH_3_), 20.4 (CH_2_), 55.4 (CH_3_), 101.6 (CH_Ar_), 111.1 (C_q_), 111.6 (CH_Ar_), 114.7 (CH_Ar_), 129.1 (2CH_Ar_), 129.9 (C_q_), 130.2 (C_q_), 131.2 (2CH_Ar_), 133.9 (C_q_), 136.1 (C_q_), 137.7 (C_q_), 155.6 (C_q_), 162.3 (C_q_S), 167.8 (C_q_), 177.8 (C_q_); HRMS (ESI-MS) m/z calcd for C_20_H_17_ClN_3_O_3_S [M+H]^+^: 414.0679, found: 414.0680;Rf (CH_2_Cl_2_/CH_3_OH = 9.8/0.2) 0.52.

#### 3.1.7. General Procedure for the Synthesis of the Nitric Oxide-Releasing Indomethacin Derivatives with 2-mercapto-1,3,4-oxadiazol Scaffold (**8a–p**)

To a suspension of (4-chlorobenzoyl)-5-methoxy-2-methyl-1*H*-indol-3-yl)methyl)- 1,3,4-oxadiazol **(7)** (1.0 Eq.) in acetonitrile (70 mL for 4 mmol scale), the corresponding (halidealkyl)phenoxy nitrates **5a–p** (1.1 Eq.) were added in one portion followed by Et_2_N being added drop wise(1.5 Eq.), according to the method reported in the literature, which was adapted to our synthesis in terms of the ratio of reagents, solvent, time of reaction and purification method [125,126]. The mixture was stirred at room temperature for 3–6 h and then the solvent was removed under reduced pressure. The residue was taken up with EtOAc (50 mL) and successively washed with distilled water (2 × 50 mL) and finally with brine (50 mL). The combined organic layers were dried over magnesium sulfate and solvent was removed under reduced pressure to obtain a crude product that was purified by flash column chromatography to produce the pure products **8a–p**.

2-(4-(((5-((1-(4-Chlorobenzoyl)-5-methoxy-2-methyl-1H-indol-3-yl)methyl)-1,3,4-oxadiazol-2-yl)thio)methyl)phenoxy)ethyl nitrate (**8a**): pale yellow solid, yield 88%, m.p. 108–110 °C; ^1^H NMR (250 MHz, DMSO-d_6_) δ = 2.29 (s, 3H), 3.74 (s, 3H), 4.19–4.28 (m, 2H), 4.36 (s, 2H), 4.37 (s, 2H), 4.79–4.91 (m, 2H), 6.74 (dd, J = 9.0 Hz, 2.5 Hz, 1H), 6.73–6.86 (m, 2H), 6.93 (dd, J = 9.0 Hz, 0.5 Hz, 1H), 7.09 (dd, J = 2.6 Hz, 0.5 Hz, 1H), 7.14–7.30 (m, 2H), 7.54–7.76 (m, 4H); ^13^C NMR (101 MHz, DMSO-d_6_) δ = 13.1 (CH_3_), 20.0 (CH_2_), 35.4 (CH_2_), 55.4 (CH_3_), 64.0 (CH_2_), 71.9 (CH_2_), 101.5 (CH_Ar_), 111.6 (CH_Ar_), 111.9 (C_q_), 114.4 (2CH_Ar_), 114.7 (CH_Ar_), 128.9 (C_q_), 129.1 (2CH_Ar_), 129.9 (C_q_), 130.3 (2CH_Ar_), 131.2 (2CH_Ar_), 133.9 (C_q_), 135.7 (C_q_), 137.8 (C_q_), 155.6 (C_q_), 157.3 (C_q_), 163.0 (C_q_), 166.1 (2C_q_), 167.9 (C_q_); HRMS (ESI-MS) m/z calcd for C_29_H_26_ClN_4_O_7_S [M+H]^+^: 609.1211, found: 609.1205; Rf (petroleum ether/EtOAc = 4/6) 0.66.

2-(4-(((5-((1-(4-Chlorobenzoyl)-5-methoxy-2-methyl-1H-indol-3-yl)methyl)-1,3,4-oxadiazol-2-yl)thio)methyl)-2-fluorophenoxy)ethyl nitrate (**8b**): pale yellow solid, yield 86%, m.p. 105–107 °C; ^1^H NMR (400 MHz, DMSO-d_6_) δ = 2.28 (s, 3H), 3.74 (s, 3H), 4.29–4.34 (m, 2H), 4.36 (s, 2H), 4.38 (s, 2H), 4.81–4.93 (m, 2H), 6.74 (dd, J = 9.0 Hz, 2.5 Hz, 1H), 6.92 (d, J = 9.0 Hz, 1H), 7.01 (t, J = 8.5 Hz, 1H), 7.07 (dd, J = 9.0 Hz, 2.0 Hz, 2H), 7.25 (dd, J = 12.2 Hz, 2.0 Hz, 1H), 7.59–7.73 (m, 4H); ^13^C NMR (101 MHz, DMSO-d_6_) δ = 13.0 (CH_3_), 20.0 (CH_2_), 34.9 (CH_2_), 55.4 (CH_3_), 65.2 (CH_2_), 71.7 (CH_2_), 101.5 (CH_Ar_), 111.6 (CH_Ar_), 111.8 (C_q_), 114.7 (CH_Ar_), 114.9 (d, J =1.8 Hz, CH_Ar_), 116.6 (CH_Ar_), 116.8 (C_q_), 125.3 (d, J = 3.4 Hz, CH_Ar_), 129.0 (2CH_Ar_), 129.9 (C_q_), 130.2 (d, J = 3.3 Hz, C_q_), 131.2 (2CH_Ar_), 133.9 (C_q_), 135.7 (C_q_), 137.7 (C_q_), 145.2 (d, J = 10.6 Hz, C_q_), 151.1 (d, J = 244.6 Hz, C_q_F), 155.6 (C_q_), 162.9 (C_q_), 166.2 (C_q_), 167.9 (C_q_); ^19^F NMR (376 MHz, DMSO-d_6_) δ = −134.2 (dd, J = 12.2 Hz, 8.6 Hz); HRMS (ESI-MS) m/z calcd for C_29_H_25_ClFN_4_O_7_S [M+H]^+^: 627.1117, found: 627.1102; Rf (petroleum ether/EtOAc = 3/7) 0.66.

2-(2-Chloro-4-(((5-((1-(4-chlorobenzoyl)-5-methoxy-2-methyl-1H-indol-3-yl)methyl)-1,3,4-oxadiazol-2-yl)thio)methyl)phenoxy)ethyl nitrate (**8c**): pale yellow solid, yield 78%, m.p. 95–97 °C; ^1^H NMR (400 MHz, DMSO-d_6_) δ = 2.28 (s, 3H), 3.73 (s, 3H), 4.29–4.37 (m, 2H), 4.35 (s, 2H), 4.38 (s, 2H), 4.84–4.92 (m, 2H), 6.74 (dd, J = 9.0 Hz, 2.5 Hz, 1H), 6.92 (d, J = 9.0 Hz, 1H), 6.99 (d, J = 8.5 Hz, 1H), 7.08 (d, J = 2.5 Hz, 1H), 7.23 (dd, J = 8.5 Hz, 2.2 Hz, 1H), 7.47 (d, J = 2.2 Hz, 1H), 7.59–7.75 (m, 4H); ^13^C NMR (101 MHz, DMSO-d_6_) δ = 13.1 (CH_3_), 20.0 (CH_2_), 34.7 (CH_2_), 55.4 (CH_3_), 65.4 (CH_2_), 71.5 (CH_2_), 101.5 (CH_Ar_), 111.6 (CH_Ar_), 111.8 (C_q_), 113.8 (CH_Ar_), 114.7 (CH_Ar_), 121.2 (C_q_), 128.9 (CH_Ar_), 129.0 (2CH_Ar_), 129.9 (C_q_), 130.3 (C_q_), 130.5 (CH_Ar_), 130.6 (C_q_), 131.2 (2CH_Ar_), 133.9 (C_q_), 135.7 (C_q_), 137.8 (C_q_), 152.6 (C_q_), 155.6 (C_q_), 162.9 (C_q_), 166.2 (C_q_), 167.9 (C_q_); HRMS (ESI-MS) m/z calcd for C_29_H_25_Cl_2_N_4_O_7_S [M+H]^+^: 643.0821, found: 643.0816; Rf (petroleum ether/EtOAc = 4/6) 0.56.

2-(2,6-Dichloro-4-(((5-((1-(4-chlorobenzoyl)-5-methoxy-2-methyl-1H-indol-3-yl)methyl)-1,3,4-oxadiazol-2-yl)thio)methyl)phenoxy)ethyl nitrate (**8d**): pale yellow solid, yield 89%, m.p. 125–127 °C; ^1^H NMR (250 MHz, DMSO-d_6_) δ = 2.28 (s, 3H), 3.73 (s, 3H), 4.22–4.32 (m, 2H), 4.35 (s, 2H), 4.42 (s, 2H), 4.80–4.91 (m, 2H), 6.72 (dd, J = 9.0 Hz, 2.5 Hz, 1H), 6.91 (dd, J = 9.0 Hz, 0.5 Hz, 1H), 7.07 (dd, J = 2.5 Hz, 0.5 Hz, 1H), 7.54 (s, 2H), 7.59–7.74 (m, 4H); ^13^C NMR (101 MHz, DMSO-d_6_) δ = 13.1 (CH_3_), 20.0 (CH_2_), 34.0 (CH_2_), 55.4 (CH_3_), 69.4 (CH_2_), 72.4 (CH_2_), 101.5 (CH_Ar_), 111.6 (CH_Ar_), 111.8 (C_q_), 114.7 (CH_Ar_), 128.0 (2C_q_), 129.0 (2CH_Ar_), 129.7 (2CH_Ar_), 129.9 (C_q_), 130.2 (C_q_), 131.2 (2CH_Ar_), 133.9 (C_q_), 135.7 (C_q_), 135.7 (C_q_), 137.7 (C_q_), 149.4 (C_q_), 155.6 (C_q_), 162.8 (C_q_), 166.3 (C_q_), 167.8 (C_q_); HRMS (ESI-MS) m/z calcd for C_29_H_24_Cl_3_N_4_O_7_S [M+H]^+^: 677.0431, found: 677.0723; Rf (petroleum ether/EtOAc = 5/5) 0.49.

2-(4-(((5-((1-(4-Chlorobenzoyl)-5-methoxy-2-methyl-1H-indol-3-yl)methyl)-1,3,4-oxadiazol-2-yl)thio)methyl)-2-nitrophenoxy)ethyl nitrate (**8e**): pale yellow solid, yield 93%, m.p. 92–94 °C;. ^1^H NMR (400 MHz, DMSO-d_6_) δ = 2.27 (s, 3H), 3.73 (s, 3H), 4.35 (s, 2H), 4.39–4.51 (m, 2H), 4.47 (s, 2H), 4.78–4.95 (m, 2H), 6.73 (dd, J = 9.0 Hz, 2.5 Hz, 1H), 6.91 (d, J = 9.0 Hz, 1H), 7.06 (d, J = 2.5 Hz, 1H), 7.24 (d, J = 8.7 Hz, 1H), 7.55–7.74 (m, 5H), 7.96 (d, J = 2.3 Hz, 1H); ^13^C NMR (101 MHz, DMSO-d_6_) δ = 13.0 (CH_3_), 20.0 (CH_2_), 34.2 (CH_2_), 55.4 (CH_3_), 66.0 (CH_2_), 71.3 (CH_2_), 101.5 (CH_Ar_), 111.6 (CH_Ar_), 111.8 (C_q_), 114.7 (CH_Ar_), 115.4 (CH_Ar_), 125.5 (CH_Ar_), 129.0 (2CH_Ar_), 129.9 (C_q_), 130.1 (C_q_), 130.2 (C_q_), 131.2 (2CH_Ar_), 133.9 (C_q_), 134.9 (CH_Ar_), 135.7 (C_q_), 137.7 (C_q_), 139.0 (C_q_), 150.0 (C_q_), 155.6 (C_q_), 162.8 (C_q_), 166.2 (C_q_), 167.9 (C_q_); HRMS (ESI-MS) m/z calcd for C_29_H_25_ClN_5_O_9_S [M+H]^+^: 654.1062, found: 654.1053; Rf (CH_2_Cl_2_/CH_3_OH = 9.8/0.2) 0.73.

2-(4-(((5-((1-(4-Chlorobenzoyl)-5-methoxy-2-methyl-1H-indol-3-yl)methyl)-1,3,4-oxadiazol-2-yl)thio)methyl)-2-methoxyphenoxy)ethyl nitrate (**8f**): pale yellow solid, yield 83%, m.p. 95–97 °C; ^1^H NMR (250 MHz, DMSO-d_6_) δ = 2.29 (s, 3H), 3.69 (s, 3H), 3.74 (s, 3H), 4.14–4.27 (m, 2H), 4.36 (s, 2H), 4.38 (s, 2H), 4.76–4.92 (m, 2H), 6.74 (dd, J = 9.0 Hz, 2.5 Hz, 1H), 6.78 (d, J = 1.1 Hz, 2H), 6.93 (dd, J = 9.0 Hz, 0.5 Hz, 1H), 7.03 (d, J = 1.2 Hz, 1H), 7.08 (d, J = 2.5 Hz, 1H), 7.58–7.73 (m, 4H); ^13^C NMR (101 MHz, DMSO-d_6_) δ = 13.1 (CH_3_), 20.0 (CH_2_), 35.9 (CH_2_), 55.4 (CH_3_), 55.4 (CH_3_), 64.9 (CH_2_), 72.0 (CH_2_), 101.5 (CH_Ar_), 111.6 (CH_Ar_), 111.9 (C_q_), 113.0 (CH_Ar_), 113.5 (CH_Ar_), 114.7 (CH_Ar_), 121.2 (CH_Ar_), 129.0 (2CH_Ar_), 129.6 (C_q_), 129.9 (C_q_), 130.3 (C_q_), 131.2 (2CH_Ar_), 133.9 (C_q_), 135.7 (C_q_), 137.7 (C_q_), 146.8 (C_q_), 148.9 (C_q_), 155.6 (C_q_), 163.1 (C_q_), 166.1 (C_q_), 167.9 (C_q_); HRMS (ESI-MS) m/z calcd for C_30_H_28_ClN_4_O_8_S [M+H]^+^: 639.1316, found: 639.1311; Rf (petroleum ether/EtOAc = 4/6) 0.66.

2-(4-(((5-((1-(4-Chlorobenzoyl)-5-methoxy-2-methyl-1H-indol-3-yl)methyl)-1,3,4-oxadiazol-2-yl)thio)methyl)-2-ethoxyphenoxy)ethyl nitrate (**8g**): pale yellow solid, yield 91%, m.p. 122–124 °C; ^1^H NMR (250 MHz, DMSO-d_6_) δ = 1.27 (t, J = 6.9 Hz, 3H), 2.29 (s, 3H), 3.74 (s, 3H), 3.94 (q, J = 6.9 Hz, 2H), 4.14–4.27 (m, 2H), 4.36 (s, 2H), 4.37 (s, 2H), 4.75–4.91 (m, 2H), 6.74 (dd, J = 9.0 Hz, 2.5 Hz, 1H), 6.79 (d, J = 1.1 Hz, 2H), 6.93 (dd, J = 9.0 Hz, 0.5 Hz, 1H), 7.02 (d, J = 1.1Hz, 1H), 7.08 (dd, J = 2.5 Hz, 0.5 Hz, 1H), 7.57–7.73 (m, 4H); ^13^C NMR (101 MHz, DMSO-d_6_) δ = 13.1 (CH_3_), 14.6 (CH_3_), 20.0 (CH_2_), 35.8 (CH_2_), 55.4 (CH_3_), 63.9 (CH_2_), 65.3 (CH_2_), 71.9 (CH_2_), 101.5 (CH_Ar_), 111.6 (CH_Ar_), 111.9 (C_q_), 114.3 (CH_Ar_), 114.5 (CH_Ar_), 114.7 (CH_Ar_), 121.3 (CH_Ar_), 129.0 (2CH_Ar_), 129.8 (C_q_), 129.9 (C_q_), 130.3 (C_q_), 131.2 (2CH_Ar_), 133.9 (C_q_), 135.7 (C_q_), 137.8 (C_q_), 147.1 (C_q_), 148.3 (C_q_), 155.6 (C_q_), 163.1 (C_q_), 166.0 (C_q_), 167.9 (C_q_); HRMS (ESI-MS) m/z calcd for C_31_H_30_ClN_4_O_8_S [M+H]^+^: 653.1473, found: 653.1459; Rf (petroleum ether/EtOAc = 4/6) 0.58.

2-(4-(((5-((1-(4-Chlorobenzoyl)-5-methoxy-2-methyl-1H-indol-3-yl)methyl)-1,3,4-oxadiazol-2-yl)thio)methyl)-2,6-dimethoxyphenoxy)ethyl nitrate (**8h**): pale yellow solid, yield 91%, m.p. 110–112 °C; ^1^H NMR (250 MHz, DMSO-d_6_) δ = 2.28 (s, 3H), 3.68 (s, 6H), 3.74 (s, 3H), 4.03–4.18 (m, 2H), 4.36 (s, 2H), 4.42 (s, 2H), 4.64–4.80 (m, 2H), 6.73 (dd, J = 9.0 Hz, 2.5 Hz, 1H), 6.73 (s, 2H), 6.92 (dd, J = 9.0 Hz, 0.5 Hz, 1H), 7.09 (d, J = 2.3 Hz, 1H), 7.57–7.74 (m, 4H); ^13^C NMR (101 MHz, DMSO-d_6_) δ = 13.1 (CH_3_), 20.0 (CH_2_), 36.3 (CH_2_), 55.4 (CH_3_), 55.8 (2CH_3_), 68.5 (CH_2_), 72.8 (CH_2_), 101.5 (CH_Ar_), 106.2 (2CH_Ar_), 111.5 (CH_Ar_), 111.8 (C_q_), 114.7 (CH_Ar_), 129.0 (2CH_Ar_), 129.9 (C_q_), 130.2 (C_q_), 131.2 (2CH_Ar_), 132.3 (C_q_), 133.9 (C_q_), 135.3 (C_q_), 135.7 (C_q_), 137.7 (C_q_), 152.7 (2C_q_), 155.6 (C_q_), 163.1 (C_q_), 166.1 (C_q_), 167.9 (C_q_); HRMS (ESI-MS) m/z calcd for C_31_H_30_ClN_4_O_9_S [M+H]^+^: 669.1422, found: 669.1411; Rf (petroleum ether/EtOAc = 4/6) 0.57.

2-(3-(((5-((1-(4-Chlorobenzoyl)-5-methoxy-2-methyl-1H-indol-3-yl)methyl)-1,3,4-oxadiazol-2-yl)thio)methyl)phenoxy)ethyl nitrate (**8i**): pale yellow solid, yield 80%, m.p. 98–100 °C; ^1^H NMR (250 MHz, DMSO-d_6_) δ = 2.28 (s, 3H), 3.74 (s, 3H), 4.15–4.31 (m, 2H), 4.35 (s, 2H), 4.41 (s, 2H), 4.76–4.93 (m, 2H), 6.73 (dd, J = 9.0 Hz, 2.5 Hz, 1H), 6.79 – 6.98 (m, 2H), 6.92 (dd, J = 9.0 Hz, 0.5 Hz, 1H), 6.98–7.03 (m, 1H), 7.08 (dd, J = 2.6 Hz, 0.5 Hz, 1H), 7.10–7.20 (m, 1H), 7.58–7.73 (m, 4H); ^13^C NMR (101 MHz, DMSO-d_6_) δ = 13.0 (CH_3_), 20.0 (CH_2_), 35.7 (CH_2_), 55.4 (CH_3_), 63.9 (CH_2_), 71.9 (CH_2_), 101.5 (CH_Ar_), 111.6 (CH_Ar_), 111.9 (C_q_), 113.6 (CH_Ar_), 114.7 (CH_Ar_), 115.3 (CH_Ar_), 121.6 (CH_Ar_), 129.1 (2CH_Ar_), 129.6 (CH_Ar_), 129.9 (C_q_), 130.2 (C_q_), 131.2 (2CH_Ar_), 133.9 (C_q_), 135.7 (C_q_), 137.7 (C_q_), 138.1 (C_q_), 155.6 (C_q_), 157.8 (C_q_), 163.0 (C_q_), 166.1 (C_q_), 167.9 (C_q_). HRMS (ESI-MS) m/z calcd for C_29_H_26_ClN_4_O_7_S [M+H]^+^: 609.1211, found: 609.1207; Rf (petroleum ether/EtOAc = 4/6) 0.63.

2-(5-(((5-((1-(4-Chlorobenzoyl)-5-methoxy-2-methyl-1H-indol-3-yl)methyl)-1,3,4-oxadiazol-2-yl)thio)methyl)-2-methoxyphenoxy)ethyl nitrate (**8j**): pale yellow solid, yield 75%, m.p. 104–106 °C; ^1^H NMR (250 MHz, DMSO-d_6_) δ = 2.28 (s, 3H), 3.70 (s, 3H), 3.74 (s, 3H), 4.10 – 4.26 (m, 2H), 4.36 (s, 4H), 4.75–4.92 (m, 2H), 6.70–6.76 (m, 1H), 6.77 (s, 1H), 6.83 (dd, J = 8.3 Hz, 1.9 Hz, 1H), 6.93 (dd, J = 9.0 Hz, 0.5 Hz, 1H), 7.06 (d, J = 1.9 Hz, 1H), 7.08 (dd, J = 2.5 Hz, 0.5 Hz, 1H), 7.59–7.74 (m, 4H); ^13^C NMR (101 MHz, DMSO-d_6_) δ = 13.1 (CH_3_), 20.0 (CH_2_), 35.8 (CH_2_), 55.4 (CH_3_), 55.5 (CH_3_), 65.0 (CH_2_), 71.9 (CH_2_), 101.5 (CH_Ar_), 111.6 (CH_Ar_), 111.9 (C_q,_ CH_Ar_), 114.7 (CH_Ar_), 114.8 (CH_Ar_), 122.3 (CH_Ar_), 128.5 (C_q_), 129.1 (2CH_Ar_), 129.9 (C_q_), 130.3 (C_q_), 131.2 (2CH_Ar_), 133.9 (C_q_), 135.7 (C_q_), 137.7 (C_q_), 147.0 (C_q_), 148.8 (C_q_), 155.6 (C_q_), 163.0 (C_q_), 166.0 (C_q_), 167.9 (C_q_); HRMS (ESI-MS) m/z calcd for C_30_H_28_ClN_4_O_8_S [M+H]^+^: 639.1316, found: 639.1313; Rf (petroleum ether/EtOAc = 4/6) 0.53.

2-(5-(((5-((1-(4-Chlorobenzoyl)-5-methoxy-2-methyl-1H-indol-3-yl)methyl)-1,3,4-oxadiazol-2-yl)thio)methyl)-2-nitrophenoxy)ethyl nitrate (**8k**): pale yellow solid, yield 88%, m.p. 90–92 °C; ^1^H NMR (400 MHz, DMSO-d_6_) δ = 2.27 (s, 3H), 3.72 (s, 3H), 4.35 (s, 2H), 4.40–4.46 (m, 2H), 4.50 (s, 2H), 4.81–4.95 (m, 2H), 6.73 (dd, J = 9.0 Hz, 2.5 Hz, 1H), 6.90 (d, J = 9.0 Hz, 1H), 7.05 (d, J = 2.5 Hz, 1H), 7.09 (dd, J = 8.3 Hz, 1.6 Hz, 1H), 7.47 (d, J = 1.6 Hz, 1H), 7.60–7.71 (m, 4H), 7.73 (d, J = 8.3 Hz, 1H); ^13^C NMR (101 MHz, DMSO-d_6_) δ = 13.0 (CH_3_), 20.0 (CH_2_), 35.1 (CH_2_), 55.4 (CH_3_), 66.0 (CH_2_), 71.2 (CH_2_), 101.5 (CH_Ar_), 111.5 (CH_Ar_), 111.8 (C_q_), 114.7 (CH_Ar_), 115.9 (CH_Ar_), 121.5 (CH_Ar_), 125.2 (CH_Ar_), 129.0 (2CH_Ar_), 129.9 (C_q_), 130.2 (C_q_), 131.2 (2CH_Ar_), 133.9 (C_q_), 135.8 (C_q_), 137.8 (C_q_), 138.6 (C_q_), 144.0 (C_q_), 150.5 (C_q_), 155.6 (C_q_), 162.6 (C_q_), 166.3 (C_q_), 167.9 (C_q_); HRMS (ESI-MS) m/z calcd for C_29_H_25_ClN_5_O_9_S [M+H]^+^: 654.1062, found: 654.1049; Rf (petroleum ether/EtOAc = 3/7) 0.48.

2-(2-Chloro-3-(((5-((1-(4-chlorobenzoyl)-5-methoxy-2-methyl-1H-indol-3-yl)methyl)-1,3,4-oxadiazol-2-yl)thio)methyl)-6-methoxyphenoxy)ethyl nitrate (**8l**): pale yellow solid, yield 87%, m.p. 117–119 °C; ^1^H NMR (400 MHz, DMSO-d_6_) δ = 2.29 (s, 3H), 3.74 (s, 3H), 3.76 (s, 3H), 4.17–4.27 (m, 2H), 4.37 (s, 2H), 4.44 (s, 2H), 4.71–4.86 (m, 2H), 6.74 (dd, J = 9.0 Hz, 2.5 Hz, 1H), 6.82 (d, J = 8.6 Hz, 1H), 6.93 (d, J = 8.9 Hz, 1H), 7.10 (dd, J = 5.5 Hz, 3.0 Hz, 2H), 7.60–7.73 (m, 4H); ^13^C NMR (101 MHz, DMSO-d_6_) δ = 13.1 (CH_3_), 20.0 (CH_2_), 34.3 (CH_2_), 55.4 (CH_3_), 56.1 (CH_3_), 68.8 (CH_2_), 72.7 (CH_2_), 101.5 (CH_Ar_), 111.0 (CH_Ar_), 111.6 (CH_Ar_), 111.8 (C_q_), 114.7 (CH_Ar_), 126.2 (C_q_), 126.4 (CH_Ar_), 127.4 (C_q_), 129.0 (2CH_Ar_), 129.9 (C_q_), 130.2 (C_q_), 131.2 (2CH_Ar_), 133.9 (C_q_), 135.8 (C_q_), 137.8 (C_q_), 143.4 (C_q_), 152.9 (C_q_), 155.6 (C_q_), 162.6 (C_q_), 166.3 (C_q_), 167.9 (C_q_); HRMS (ESI-MS) m/z calcd for C_30_H_27_Cl_2_N_4_O_8_S [M+H]^+^: 673.0927, found: 673.0918; Rf (petroleum ether/EtOAc = 4/6) 0.53.

2-(2-Bromo-3-(((5-((1-(4-chlorobenzoyl)-5-methoxy-2-methyl-1H-indol-3-yl)methyl)-1,3,4-oxadiazol-2-yl)thio)methyl)-6-methoxyphenoxy)ethyl nitrate (**8m**): pale yellow solid, yield 88%, m.p. 94–96 °C; ^1^H NMR (400 MHz, DMSO-d_6_) δ = 2.29 (s, 3H), 3.74 (s, 3H), 3.77 (s, 3H), 4.16–4.24 (m, 2H), 4.37 (s, 2H), 4.46 (s, 2H), 4.75–4.87 (m, 2H), 6.74 (dd, J = 9.0 Hz, 2.5 Hz, 1H), 6.86 (d, J = 8.7 Hz, 1H), 6.93 (d, J = 8.9 Hz, 1H), 7.10 (d, J = 2.5 Hz, 1H), 7.12 (d, J = 8.6 Hz, 1H), 7.60–7.72 (m, 4H); ^13^C NMR (101 MHz, DMSO-d_6_) δ = 13.1 (CH_3_), 20.0 (CH_2_), 36.9 (CH_2_), 55.4 (CH_3_), 56.1 (CH_3_), 68.6 (CH_2_), 72.7 (CH_2_), 101.5 (CH_Ar_), 111.6 (CH_Ar_), 111.7 (CH_Ar_), 111.9 (C_q_), 114.7 (CH_Ar_), 119.2 (C_q_), 126.6 (CH_Ar_), 127.7 (C_q_), 129.0 (2CH_Ar_), 129.9 (C_q_), 130.3 (C_q_), 131.2 (2CH_Ar_), 133.9 (C_q_), 135.8 (C_q_), 137.8 (C_q_), 144.5 (C_q_), 152.7 (C_q_), 155.6 (C_q_), 162.6 (C_q_), 166.3 (C_q_), 167.9 (C_q_); HRMS (ESI-MS) m/z calcd for C_30_H_27_BrClN_4_O_8_S [M+H]^+^: 717.0421, found: 717.0409; Rf (petroleum ether/EtOAc = 4/6) 0.55.

2-(2-(((5-((1-(4-Chlorobenzoyl)-5-methoxy-2-methyl-1H-indol-3-yl)methyl)-1,3,4-oxadiazol-2-yl)thio)methyl)phenoxy)ethyl nitrate (**8n**): pale yellow solid, yield 70%, m.p. 89–91 °C; ^1^H NMR (400 MHz, DMSO-d_6_) δ = 2.28 (s, 3H), 3.74 (s, 3H), 4.28–4.40 (m, 2H), 4.34 (s, 2H), 4.35 (s, 2H), 4.79–4.98 (m, 2H), 6.74 (dd, J = 8.9 Hz, 2.5 Hz, 1H), 6.78 (td, J = 7.5 Hz, 1.0 Hz, 1H), 6.92 (d, J = 8.9 Hz, 1H), 7.01 (dd, J = 8.4 Hz, 1.0 Hz, 1H), 7.08 (d, J = 2.5 Hz, 1H), 7.19 (dd, J = 7.5 Hz, 1.7 Hz, 1H), 7.25 (ddd, J = 8.2 Hz, 7.4 Hz, 1.7 Hz, 1H), 7.58–7.72 (m, 4H); ^13^C NMR (101 MHz, DMSO-d_6_) δ = 13.1 (CH_3_), 20.0 (CH_2_), 31.4 (CH_2_), 55.4 (CH_3_), 64.7 (CH_2_), 71.8 (CH_2_), 101.5 (CH_Ar_), 111.6 (CH_Ar_), 111.9 (C_q_), 112.2 (CH_Ar_), 114.7 (CH_Ar_), 120.8 (CH_Ar_), 124.3 (C_q_), 129.0 (2CH_Ar_), 129.6 (CH_Ar_), 129.9 (C_q_), 130.3 (C_q_), 130.3 (CH_Ar_), 131.2 (2CH_Ar_), 133.9 (C_q_), 135.7 (C_q_), 137.7 (C_q_), 155.6 (C_q_), 155.7 (C_q_), 163.3 (C_q_), 166.1 (C_q_), 167.9 (C_q_); HRMS (ESI-MS) m/z calcd for C_29_H_26_ClN_4_O_7_S [M+H]^+^: 609.1211, found: 609.1195; Rf (petroleum ether/EtOAc = 4/6) 0.68.

2-(4-Bromo-2-(((5-((1-(4-chlorobenzoyl)-5-methoxy-2-methyl-1H-indol-3-yl)methyl)-1,3,4-oxadiazol-2-yl)thio)methyl)phenoxy)ethyl nitrate (**8o**): pale yellow solid, yield 87%, m.p. 91–93 °C; ^1^H NMR (400 MHz, DMSO-d_6_) δ = 2.28 (s, 3H), 3.74 (s, 3H), 4.26–4.41 (m, 2H), 4.34 (s, 4H), 4.79–4.92 (m, 2H), 6.73 (dd, J = 8.9 Hz, 2.5 Hz, 1H), 6.92 (d, J = 8.9 Hz, 1H), 7.00 (d, J = 8.8 Hz, 1H), 7.07 (d, J = 2.5 Hz, 1H), 7.44 (dd, J = 8.7 Hz, 2.5 Hz, 1H), 7.50 (d, J = 2.5 Hz, 1H), 7.59–7.72 (m, 4H); ^13^C NMR (101 MHz, DMSO-d_6_) δ = 13.1 (CH_3_), 20.0 (CH_2_), 30.7 (CH_2_), 55.4 (CH_3_), 65.1 (CH_2_), 71.7 (CH_2_), 101.4 (CH_Ar_), 111.6 (CH_Ar_), 111.8 (C_q_), 112.0 (C_q_), 114.4 (CH_Ar_), 114.7 (CH_Ar_), 127.1 (C_q_), 129.0 (2CH_Ar_), 129.9 (C_q_), 130.2 (C_q_), 131.2 (2CH_Ar_), 131.9 (CH_Ar_), 132.7 (CH_Ar_), 133.9 (C_q_), 135.7 (C_q_), 137.7 (C_q_), 155.0 (C_q_), 155.6 (C_q_), 163.1 (C_q_), 166.2 (C_q_), 167.8 (C_q_); HRMS (ESI-MS) m/z calcd for C_29_H_25_BrClN_4_O_7_S [M+H]^+^: 687.0316, found: 687.0302; Rf (petroleum ether/EtOAc = 4/6) 0.70.

2-(4-(2-((5-((1-(4-Chlorobenzoyl)-5-methoxy-2-methyl-1H-indol-3-yl)methyl)-1,3,4-oxadiazol-2-yl)thio)ethyl)phenoxy)ethyl nitrate (**8p**): yellow solid, yield 82%, m.p. 92–94 °C; ^1^H NMR (250 MHz, DMSO-d_6_) δ = 2.29 (s, 3H), 2.91 (t, J = 7.5 Hz, 2H), 3.25–3.45 (m, 2H), 3.74 (s, 3H), 4.18–4.31 (m, 2H), 4.36 (s, 2H), 4.79–4.93 (m, 2H), 6.74 (dd, J = 9.0 Hz, 2.5 Hz, 1H), 6.81–6.89 (m, 2H), 6.92 (dd, J = 9.0 Hz, 0.5 Hz, 1H), 7.04–7.16 (m, 3H), 7.56–7.73 (m, 4H); ^13^C NMR (101 MHz, DMSO-d_6_) δ = 13.1 (CH_3_), 20.0 (CH_2_), 33.4 (CH_2_), 34.0 (CH_2_), 55.4 (CH_3_), 63.9 (CH_2_), 72.0 (CH_2_), 101.5 (CH_Ar_), 111.5 (CH_Ar_), 111.9 (C_q_), 114.4 (2CH_Ar_), 114.7 (CH_Ar_), 129.0 (2CH_Ar_), 129.7 (2CH_Ar_), 129.9 (C_q_), 130.2 (C_q_), 131.2 (2CH_Ar_), 131.7 (C_q_), 133.9 (C_q_), 135.7 (C_q_), 137.7 (C_q_), 155.6 (C_q_), 156.5 (C_q_), 163.4 (C_q_), 165.9 (C_q_), 167.9 (C_q_); HRMS (ESI-MS) m/z calcd for C_30_H_28_ClN_4_O_7_S [M+H]^+^: 623.1367, found: 623.1364; Rf (petroleum ether/EtOAc = 5/5) 0.63.

### 3.2. In Silico Docking Study

In order to study the selectivity of the new NO-IND-OXDs **8a–p** for COX isoenzymes (COX-1 and COX-2), AutoDock 4.2.6 software was used [82,127,128,129]. The results, expressed as docking score, were compared to IND, DCF and CCB, used as reference drugs.

#### 3.2.1. Generating the Receptor Coordinate File (RCF)

X-ray crystallographic structures for COX-1 (pdb code: 4o1z) and COX-2 (pdb code: 3nt1)) used in this research were selected in terms of the quality of the model obtained from RCSB Protein Data Bank and were processed prior to docking. From downloaded experimental structures, the following were removed: the ligands, the water molecules, cofactors and ions that should not be included in the receptor with a text editor. After that, by reading the coordinates, adding charges, merging nonpolar hydrogens and assigning appropriate atom types, each receptor was converted to PDBQT format file using AutoDock 4.2.6.

#### 3.2.2. Generating the Ligand Coordinate File (LCF)

The PDB coordinate files of compounds **8a–p** were generated using Chimera 1.14 software. The LCF describes the ligand structure through several types of records (such as ROOT, ENDROOT, BRANCH, ENDBRANCH and TORSDOF) that are recognized by AutoDock. Each structure was energy minimized and converted to PDBQT format file. To study the interaction between a single ligand with a single receptor, with explicit calculation of affinity maps, a docking method was used. The receptors were kept rigid and the ligands were allowed to be flexible.

#### 3.2.3. Preparing the Grid Parameter File (GPF)

The GPF specifies the parameters for generating the atomic affinity maps and the PDBQT files for the receptor. For COX-2 we used a grid box of 74 × 72 × 86 points with a spacing of 0.375 Å between grid points and the grid box center was put on x = –37.882, y = –50.853 and z = –21.24. COX-1 was enclosed in a 73 × 78 × 82 grid box with 0.375 Å spacing and 251.00, 104.00 and 1.364 as x, y and z center. The binding site of COX-1 and COX-2 ligands is respectively identified by using protein visualization software such as PyMOL, DS visualizer and Chimera 1.14.

#### 3.2.4. Preparing the Docking Parameter File (DPF)

To prepare the DPF file, different parameters were selected: the ligand molecule to dock and its center and number of torsions and how many runs as well as the grid map files and the docking algorithm to use. For conformation search, the Lamarckian genetic algorithm (LGA) was applied, with the following parameters: number of runs for each docking procedure (200), number of individuals in the population (300), the maximum number of 27,000 generations simulated during each LGA run, the maximum number of evaluations at 25,000,000, a mutation rate of 0.02 and a cross over rate of 0.80, while the remaining docking parameters were set to default.

To identify the COX selectivity of ligands **8a–p**, AutoDock 4.2.6 software was applied. After performing molecular docking simulation, the best ligand molecules were evaluated on the basis of their docking score against the COX receptors. The dockings experiments were clustered with a root mean square deviation (RMSD) of 0.5 Å and were evaluated by PyMOL software. Finally, the most energetically favored orientations were selected for subsequent study.

### 3.3. In Vitro Radical Scavenging Assay

The antioxidant activity, as radical scavenging effect, of the new NO-IND-OXDs **8a**–**p** was evaluated using 2,2-diphenyl-1-picrylhydrazyl radical (DPPH) assay with slight modification [130,131,132]. The results were analyzed using IND and ASP as reference drugs and vitamin C as standard antioxidant.

*Preparation of DPPH and test solutions*. A weighed amount of DPPH (29.8 mg, 0.076 mmol) was dissolved by sonication in 50 mL methanol of analytical grade. After 30 min at darkness, a sample of 10 mL was taken and made up to 100 mL with methanol. The resulting DPPH solution (151.14 μM) was stored in the darkness at room temperature and used up on the day of preparation. The stock solutions (2600 μM) of tested derivatives **8a–p** were prepared in DMSO, then serially diluted with methanol to obtain different concentrations (2600 μM, 1500 μM, 700 μM, 620 μM, 530 μM, 440 μM, 350 μM, 260 μM and 120 μM). In the same manner with tested derivatives **8a–p**, the serially diluted solutions of IND and ASP were prepared. By dilution of vitamin C, freshly prepared solution (2612 μM) with methanol, serially diluted solutions (152.3 μM, 133.1 μM, 112.4 μM, 96.1 μM, 75.3 μM, 55.1 μM, 40.1 μM, 20.5 μM) were also prepared.

*DPPH assay procedure*. First, 500 μL from each sample of the tested compounds **8a–p**, reference drugs (IND, ASP) and vitamin C was added to 1000 μL of DPPH solution. Two blanks (blank 1: 500 μL of methanol and 1000 μL of DPPH and blank 2: 1500 μL of methanol) were also used. The mixture was kept for 3 h in the darkness at room temperature and then a 270 μL aliquot of each sample tube was added in a 96-well plate. The absorbance was then measured at 517 nm using Tecan Sunrise Remote Microplate Reader TW/ML-Abbott F039306. All tests were performed in quadruplicate.

To calculate the DPPH radical-inhibiting capacity (scavenging activity, %), the following formula was used [133,134]:Inhibition (Scavenging activity) % = [(A_CS_ − A_s_)/A_CS_] × 100(1)
where: A_s_ is the difference between the absorbance of tested sample and blank 2 and A_cs_ is the difference between absorbance of blank 1 and blank 2.

The IC_50_ (f_(x)_ = 50) of each tested compound was calculated by plotting the inhibition ratios (f_(x)_) against the sample concentration (x). The results for each experiment were represented by a dose–response curve and two types of regression lines (f(x)) were used: a sigmoid curve and a quadratic line.

The DPPH radical scavenging activity of each tested compound was also expressed as the vitamin C equivalent antioxidant capacity (CEAC), which was calculated using the following formula [135,136]:CEAC = IC_50(vit_. _C)_/IC_50(sample)_(2)
where: IC_50(sample)_ is concentration of tested compounds **8a–p** and reference drugs, respectively, necessary for 50% inhibition and IC_50(vit_. _C)_ is the concentration of vitamin C needed for 50% inhibition. The higher CEAC value means higher DPPH radical scavenging activity.

### 3.4. Thermal Denaturation of Serum Proteins

The anti-inflammatory effects of the new NO-IND-OXZs **8a–p** were predicted by a modified Mizushima’s test [87,88]. The test consists of the denaturing effect on specific proteins and assures a significant correlation between the in vitro and in vivo effects. A solution of bovine serum albumin (BSA) 0.2% in 0.9% NaCl/DMSO = 6/4 was used. Two controls (positive and negative), as well as reference drugs (IND, ASP), were also used. The positive control consisted of the action of 0.1M hydrochloric acid, as denaturing agent, in 0.9% NaCl, on the 0.2% BSA solution while the negative control consisted of untreated 0.2% BSA solution. Each sample (**8a–p**, IND, ASP) was tested at different concentrations (20 μM, 50 μM and 100 μM). The test samples and controls were incubated at 38 °C for 5 h. The degree of denaturation of BSA was quantified based on absorbance value, measured at 450 nm, using Tecan Sunrise Remote Microplate Reader TW/ML-Abbott F039306. All tests were performed in quadruplicate. The maximum value of the absorbance at 450 nm of the positive control was considered as the 100% effect. The results were expressed as averages of the percentage values (% effect) and were plotted versus the concentration of tested sample.

### 3.5. In Vitro Nitric Oxide Release Measurement

To assess the capacity of new NO-IND-OXDs **8a–p** to to release the nitric oxide (NO), the Griess colorimetric method was applied [137,138]. The Griess reagents consist of 0.34% (wt/v) *N*-(1-naphthyl)ethylenediamine (NED) solution in DMSO, 3.4% (wt/v*)* sulfanilamide (SULF) in 10% (wt/v) phosphoric acid and a mixture between 3.4% (wt/v) SULF and 1% (wt/v) mercuric chloride (SULF-HgCl_2_) in 10% (wt/v*)* phosphoric acid. The NO released from the sample was spontaneously oxidized to NO_2_^−^, which subsequently reacted with the Griess reagents to form an azo dye. As reference NO donors, *S*-nitroso-*N*-acetyl-penicillamine (SNAP), sodium nitropruside (SNP) and nitroglycerine (NTG) were used.

The experiments were performed in neutral (phosphate buffer solution—PBS) and acidic (hydrochloric acid solution—HCl) experimental conditions, in the presence or absence of L-glutathione (GSH): PBS (pH 7.5), PBS-GSH (pH 7.51), HCl (pH 1.55), HCl-GSH (pH 1.56).


*Preparation of sodium nitrite and test solutions*


A fresh sodium nitrite stock aqueous solution (0.1 M) was prepared and was standardized according to the procedure reported in the European Pharmacopoeia (real molarity = 0.0998, molarity factor = 0.998) [139]. Then, the stock solution was serially diluted with MeOH/H_2_O = 1/1 (v/v) mixture obtaining different concentrations (100 μM, 50 μM, 25 μM, 12.50 μM, 6.25 μM, 3.125 μM, 1.56 μM and 0.78 μM). The tested compounds (**8a–p**) and reference NO donors (SNAP, SNP, NTG) were dissolved in DMSO and water, respectively, to afford a stock solution of 2600 μM.


*The preparation of nitrite standard curve*


An aliquot of 170 μL sodium nitrite solution (in the range of 0.78–100 μM) was added to 50 μL solution of SULF in a 96-well plate. After 10 min, 50 μL of NED solution was added and then after another 20 min, the absorbance at 540 nm of the formed pink-red azo dye was measured. A blank sample (containing 170 μL PBS, 50 μL SULF and 50 μL NED) was prepared under similar conditions. All tests were performed in quintuplicate and the average absorbance was calculated. The calibration curve was constructed by graphical representation of the mean absorbance value in relation to the corresponding concentration of sodium nitrite.


*NO release assay*


A solution of 100 μM NO-IND-OXDs **8a–p** and reference NO donors (SNAP, SNP, NTG) was prepared by diluting 80 μL of each stock solution (2600 μM) with 2 mL of PBS, PBS-GSH, HCl and HCl-GSH. These above solutions were kept at 37–38 °C for 120 min; after that, an aliquot of 170 μL of each solution was measured and added to 50 μL of SULF and SULF-HgCl_2_, respectively, in a 96-well plate. After 10 min, 50 μL of NED solution was added and after another 20 min, the absorbance at 540 nm of the formed pink-red azo dye was measured. A blank sample (containing 170 μL PBS/PBS-GSH/HCl/HCl-GSH, 50 μL SULF/SULF-HgCl_2_ and 50 μL NED) was prepared under similar conditions. All tests were performed in quadruplicate. The percentage (%) of NO release was calculated using the following formula [137]:% NO = (C_f NO_ × 100)/C_t NO_(3)
where: C_fNO_ is the found concentration and C_tNO_ is the theoretical concentration of NO (μM).

### 3.6. Statistical Analysis

The results were expressed as mean value ± standard deviation (SD) and the analysis was performed using IBM SPSS Statistics 23 for Windows. The statistical significance of the results was assessed using the one-way and two-way analysis of variance (ANOVA’s test). To compare the differences among samples, the Tukey’s HSD test was used. A *p* value less than 0.05 was considered statistically significant.

## 4. Conclusions

Based on the beneficial effects of 1,3,4-oxadiazole-2-thiol scaffold and NO, new nitric oxide-releasing indomethacin derivatives with 1,3,4-oxadiazole-2-thiol scaffold (NO-IND-OXDs) have been synthesized. It is known that NO is an important endogenous molecule, having a critical role in protecting the GI mucosa while 1,3,4-oxadiazole-2-thiol scaffold is associated with important pharmacological effects, including anti-inflammatory and antioxidant ones. The molecular docking study revealed that **8k**, **8l** and **8m** are COX-2 inhibitors, more selective than celecoxib. It was noted that COX-2 selectivity is influenced by the position of nitrate ester moiety, *meta* and *para* being more favorable than *ortho*, as well as by the nature of the substituents from phenoxy-ethylnitrate moiety, the better influence being associated with halogens (F, Cl, Br) and NO_2_. The tested compounds also showed improved radical scavenging as well as anti-inflammatory properties, the last one measured as albumin denaturation effect. While for IND and ASP, a denaturation effect of less than 5% was recorded, all the tested compounds showed a denaturation effect ranging between 20% and 40% (at 100 µM). The NO release capacity of tested compounds is strongly influenced by the position of nitrate ester moiety and by the nature of the substituents from phenoxy-ethylnitrate moiety, being in good agreement with the COX-selectivity. The results of our study, related to COX-2 in silico interaction, DPPH^•^ radical-scavenging capacity, PBS denaturation-promoting effect and NO-releasing property, encourage us to continue our research with in vivo inflammation model assay to prove the potential effect of NO-IND-OXDs as multitarget strategy.

## Data Availability

Samples of the compounds are not available from the authors.

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
