# Peer review of "Design, Synthesis, In Silico and In Vitro Studies for New Nitric Oxide-Releasing Indomethacin Derivatives with 1,3,4-Oxadiazole-2-thiol Scaffold"

_ijms, 2021, doi:10.3390/ijms22137079_

Round 1

Reviewer 1 Report

The authors prepared a very interesting article entitled “Design, synthesis, in silico and in vitro studies for new nitric oxide-releasing indomethacin derivatives with 1,3,4-oxadioazole-2-tiol scaffold”, but they described some the known compounds as new and the HMRS analysis, in which the positive ion is not analyzed, what is unclear. Please explain at the chemistry documentation section in detail and studied database on the molecules.

A number of another errors are listed below:

At line 22 is: … indometacin … .

Should be: … indomethacin … .

Comment: see line 3.

At line 27 definition of COX is required.

Comment: The abstract is an independent part of the publication.

At line 28 is: … andantioxidant … .

Should be: … and antioxidant … .

Comment: please add a space.

At line 28 is: … in vitro … .

Should be: … in vitro … .

Comment: please write in Italic style such as in lines 2, 108, 231, 266, 310, etc.

Similar mistake to correction is in lines 1291, 1340, 1342, 1361, and 1437.

At line 29 definition of COX-2 is required, or an explanation to help the reader understand the topic.

Comment: The abstract is an independent part of the publication.

At lines 29 and 32 at the abstract molecules is defined by number of compounds 8k, 8l, 8m and 8c, 8h, 8i, 8m, 8o, respectively, but abstract is an independent part of the publication, an at CA database the reader will not see decoder at Scheme 1. Please add a brief information on the structural integrity of specific molecules mentioned to refine the general information from line 24, such as steric hindrance, electronic nature of substituents, number and position of substituents, etc., if it is possible.

At line 32 is: … compounds8c,  … .

Should be: … compounds 8c,  … .

Comment: please add a space.

At line 51 is: … 1999-2012 ... .

Should be: … 1999–2012 ... .

At line 71 is: … to inhibitthe … .

Should be: … to inhibit the … .

Comment: please add a space.

At line 73 is: … andNO … .

Should be: … and NO … .

Comment: please add a space.

At line 90 is: … core have … .

It should be much better: … core, generally 1,3,4-oxadiazole motif, have … .

At line 92 “[56] ,” there is a space to be removed before the comma.

At line 119 there is a dot mark ‘ . ‘ at the end of the paragraph that is incorrectly written in bold style.

At line 114 is: … (1a-o) … .

Should be better: … 1a-o … .

Comment: After the general name or without the name, the numbering may not be given in parentheses. Please revise in all other cases where necessary, such as in lines such 116, 118, 119, 129, 130, 132 (twice), 136 (two from the end), 140, 144, …, 398 (on the right side of the line), 403, 529 (see line 418), 539, 649, 665, 796, 990, 1014, etc.

At line 115 is: … (2) … .

Should be: … (2p) … .

Comment: Serious mistake.

On line 119 at the end of the paragraph there is an erroneously bold dot character ( . ), moreover, there is no reference to the original literature on the reduction of aldehydes with NaBH4. Please correct it.

At lines 120–123 at Scheme 1 are the numbers of the molecules that should be written in bold.

At lines 125–128 are: … 18-26h … 12-24h … 8-10h … 20-30 … 0-5 … 3-6h … .

Should be: … 18–26h … 12–24h … 8–10h … 20–30 … 0–5 … 3–6h … .

At line 127 is number of molecules: … 5a-p … .

Comment: particle numbers should be written in bold style: … 5a-p … , see line 132. 

At line 130 is: … were obtained, … .

Comment: The reaction is not new, please add refer to original source literature.

At line 132 described reactions are not new, please refer to original source literature.

At line 135 is: … -1H- … .

Should be: … -1H- … .

Comment: Recommendation in accordance with the IUPAC nomenclature.

At line 137: in general, reference should be made to the reference literature for the described reactions - they may be original reviews, if available.

At line 142 is: … 4.7 – 4.9 ppm … .

Should be: … 4.7–4.9 ppm … .

Comment: two space delete please, see line 148.

At line 143 is: … 3.7 – 3.90 ppm … .

Should be: … 3.7–3.9 ppm … .

Comment: two space delete please, see line 148.

At line 145 is: … 72-73 ppm … .

Should be: … 72–73 ppm … .

Comment: see line 148.

At line 146 is: … δ = 31-32 ppm) … .

Should be: … δ = 31–32 ppm) … .

Comment: see line 148.

At line 147 is: … δ = 4.7-4.9 ppm) … .

Should be: … δ = 4.7–4.9 ppm) … .

Comment: see line 148.

At line 149 is: … 71-72 ppm … .

Should be: … 71–72 ppm … .

Comment: see line 148.

At line 150 is: … 34-36 ppm … .

Should be: … 34–36 ppm … .

Comment: see line 148.

At line 151 is: … 162-163 ppm … .

Should be: … 162–163 ppm … .

Comment: see line 148.

At lines 181–183 in Table 1 maybe it would be better to use mathematical subtraction signs ( − ).

At line 380 is: … 40-70 mM … .

Should be: … 40–70 mM … .

Comment: see line 148.

At line 381 is: … (230-240 mesh) … . Please check if the size is correct. Perhaps it should be: … (230–400 mesh) … .

Comment: see lines such 1394, 1401, 1413, 1428, 1431, and 1435.

At line 386 is: … DMSO-d6. … .

Should be: … DMSO-d6. … .

Comment: see lines such 420 and 422.

At line 386 is: … d6.The … .

Comment: a space should be inserted after the dot mark ( . ) also at lines 393, 409, 1002, 1023, 1091,  .

At line 389 is: … 2D-RMN … .

Shouldn't it be better: … 2D-NMR … .

Comment: English please.

At line 398 is: … intermediaries … .

Comment: classically, there should be better: ... intermediates ... . Similarly please correct at lines 136, and 138.

At line 401 is: … [78–80],and … .

Should be: … … [78–80], and … .

Comment: a space should be inserted after the comma ( , ) .

At line 405 is: … 12-24 h … .

Should be: … 12–24 h … .

Comment: see lines such 148 and 406.

At line 414 is: … (3 x 50 mL) … .

Should be: … (3 × 50 mL) … .

Comment: please use mathematic multiplication mark ‘×’ . See line 378.

At line 417 is: … ether/ EtOAc … .

Should be better: … ether/EtOAc … .

Comment: see line such 419.

At lines 419–424 the authors misspelled the known molecule, ascribing it to themselves as new. The CAS Number is 38459-72-2. 1H NMR is known and Authors must be cited the source literature at the title such US2017/37008, A1 or Chemical and Pharmaceutical Bulletin 2007, 55(7), 1053–1059.

At lines 452–458 the authors misspelled the known molecule, ascribing it to themselves as new. The CAS Number is 98688-19-8 Literature m.p. is 84–77°C [J. Am. Chem. Soc. 1985, 107(24), 6993–6996]. Literature m.p. must be cited.

At lines 459–466 the authors misspelled the known molecule, ascribing it to themselves as new. Literature m.p. is 51–53°C [Chem. Abstr., 1979, 91(157445)]. Literature m.p. must be cited. Please check all the compunds described.

At line 419 is: … m.p. 73-75°C … .

Should be: … m.p. 73–75°C … .

Comment: see lines such 148 and 406.

Similar errors to correction are in lines 453, 459, 468, 495, 502, 522, 540, 568, 575, 582, 590, 602, 615, 622, 642, 680, 686, 691, 697, 703, 710, 728, 757, 776, 798, 810, 822, 834, 843, 855, 866, 878, 888, 900, 910, 922, 933, 943, 956, and 966.

At line 423 for compound 3a details is: … HRMS (EI-MS) m/z calcd for C9H11BrO2: 229.9942 [M+H]+ … .

Comment: The calculated data given for HRMS is for C9H11BrO2 molecule, but for C9H12BrO2 must be 231,0015.

Please add all HRMS analyses for Supplementary Information, alternatively elementary analyses for carbon and hydrogen and/or heteroatom is needed.

.

At lines 440–441 is: … m.p. 665-67 °C … .

Comment: please correct.

At lines 551, 1087 please add dot mark ‘ . ‘ at the end.

At line 535 is: … (2 x 50 mL) … .

Should be: … (2 × 50 mL) … .

Comment: please use mathematic multiplication mark ‘×’ . See line 378.

At line 649 is: … nitrates(4a-o) … .

Should be: … nitrates 4a-o … .

Comment: also space please add.

649

At line 650 is: … mLfor … .

Should be: … mL for … .

Comment: space mark please add.

At lines 805–806 for compound 8a details is: … HRMS (EI-MS) m/z calcd for C29H25ClN4O7S: 608.1132 [M+H]+, … .

Comment: The calculated HRMS is for C29H25ClN4O7S molecule, but for C29H26ClN4O7S+ must be 609.1205.

Please add all HRMS analyses for Supplementary Information, alternatively elementary analyses for carbon and hydrogen and/or heteroatom is needed.

At lines 818–819 is: … HRMS (EI-MS) m/z calcd for C29H24ClFN4O7S: 626.1038 [M+H]+ … .

Comment: There is calculated HRMS for C29H24ClFN4O7S molecule of 8b, but for C29H25ClFN4O7S+ of M+H+ must be 627.1111.

Please add all HRMS analyses for Supplementary Information, alternatively elementary analyses for carbon and hydrogen and/or heteroatom is needed.

At line 658 is: … (4p) … .

Should be: … (4p) … .

Comment: see line such 657.

At line 655 is: … 20-30 … .

Should be: … 20–30 … .

Comment: see lines such 148 and 406.

At line 656 is: … (2 x 50 mL) … .

Should be: … (2 × 50 mL) … .

Comment: please use mathematic multiplication mark ‘×’ . See line 378.

At line 773 is: … (3 x 50 mL) … .

Should be: … (3 × 50 mL) … .

Comment: please use mathematic multiplication mark ‘×’ . See line 378.

At line 788 is: … (1.1 … .

Should be: … (1.1 … .  

Comment: The brace opening character is in bold “ ( “ and should be in the normal style “ ( “. Please correct also at line 786.

At line 790 is: … ture,which … .

Should be: … ture, which … .

Comment: space mark please add. 

At line 792 is: … 3-6 h … .

Should be: … 3–6 h … .

Comment: see lines such 148 and 406.

At line 793 is: … (2 x 50 mL) … .

Should be: … (2 × 50 mL) … .

Comment: please use mathematic multiplication mark ‘×’ . See line 378.

At line 999 is: … box of 74x72x86 points … .

Maybe should be better: … box of 74×72×86 points … .

Comment: please use twice mathematic multiplication mark such ‘×’ . See line 378.

At lines 1000–1001 missing mathematical subtraction signs.

At line 1001 is: … in a 73x78x82 grid box … .

Maybe should be better: … in a 73×78×82 grid box … .

Comment: please use twice mathematic multiplication marks such ‘×’ . See line 378.

At lines 1030, 1032, 1054, and 1094 is: … (8a-p) … .

Should be: … 8a-p … .

Comment: Please omit the brackets and numbers in full please write in bold.

In equation (1) between lines 1044 and 1045, use the mathematical multiplication sign “ × ” if necessary.

On line 1059, please put a space in front of the word "was".

At line 1080 is: … NO2-, … .

Should be: … NO2, … .

Comment: Please insert a mathematical subtraction sign “ ”, or a valence related sign, of course in superscript.

At line 1082 the symbol of the nitrogen atom “ N “ in the systematic name should be in italics.

At line 1097 is: … 0.78-100 μM) … .

Should be: … 0.78–100 μM) … .

Comment: see lines such 148 and 406.

At line 1106 please add a forgotten space before the last bracket.

At line 1108 is: … 37-38°C … .

Should be: … 37–38°C … .

Comment: see lines such 148 and 406.

At line 1171, ref 4 – please complete the information about the article.

At line 1220, ref. 25, is: … 271-277 … .

Should be: … 271–277 … .

Comment: see lines such 1216, 1217, 1222.

At line 1265, ref. 45 please add article number.

At line 1384 please delete the mark ‘ § ‘at the end of the title.

At line 1399, ref. 95 is needed abbreviation of name Monatshefte f?r Chemie/Chemical Mon. such Monatshefte für Chemie .

At line 1407, ref. 99 is: … S. V. … .

Should be: … S.V. … .

Comment: space mark delete please.

At lines 1433–1434, ref. 108 is: … SHIMAMURA, T.; SUMIKURA, Y.; YAMAZAKI, T.; TADA, A.; KASHIWAGI, T.; ISHIKAWA, H.; MATSUI, T.; SUGIMOTO, N.; AKIYAMA, H.; UKEDA, H. … .

Should be: … Shimamura, T.; Sumikura, Y.; Yamazaki, T.; Tada, A.; Kashiwagi, T.; Ishikawa, H.; Matsui, T.; Sugimoto, N.; Akiyama, H.; Ukeda, H. … .

Comment: please use MDPI style.

At line 1435, ref. 108 is: … Applicability of the DPPH Assay for Evaluating the Antioxidant Capacity of Food Additives ^|^ndash; Inter-laboratory Evaluation Study ^|^ndash … .  

Should be better: … Applicability of the DPPH assay for evaluating the antioxidant capacity of food additives – inter-laboratory evaluation study –. … .

Comment: Please English mark use only.

At line 1444, ref. 112 is: … Prod.1998 … .

Should be: … Prod. 1998 … .

Comment: Space mark please add.

At line 1446, ref. 113 is: … Chem.2002 … .

Should be: … Chem. 2002 … .

Comment: Space mark please add.

At line 1447, ref. 114 is: … Yu, L. (Lucy) … .

It should probably be: … Yu, L.(L.) … .

At line 1448, ref. 114 is: … Chem.2006 … .

Should be: … Chem. 2006 … .

Comment: Space mark please add. 393

At line 1451, ref. 115 is: … Chem.2009 … .

Should be: … Chem. 2009 … .

Comment: Space mark please add.

At line 1453, ref. 116 is: … Chem.2017 … .

Should be: … Chem. 2017 … .

Comment: Space mark please add.

At line 1455, ref. 117 please continue, see line 1250.

Author Response

Dear Reviewer,

We are very grateful for your expertise and valuable evaluation of the quality of our manuscript and for your important and constructive suggestions. The corrections were highlighted in red in the manuscript using track changes. Please take in consideration our response to your comments/suggestions.

Thank you a lot for your expertise, your time and your availability!

In behalf of the authors,

Prof. dr. Lenuta Profire

Reviewer 2 Report

The reviewed manuscript takes up an important topic of searching for multi-target compounds among analogues of currently used non-steroidal anti-inflammatory drugs. The additional effects of the synthesized compounds may reduce the side effects of the use of analgesics. The choice of indomethacin as the target of modification is correct because it is one of the most potent anti-inflammatory drugs and at the same time one of the most toxic. The synthesis of indomethacin oxadiazole derivatives is deliberate, as shown by the results of biological and computational studies obtained by the authors. The work is clearly composed. Schemes and figures are shown correctly. The research methodology is appropriate and the conclusions are correct. 

Author Response

Dear Reviewer,

We are very grateful for your expertise and valuable evaluation of the quality of our manuscript

Prof. dr. Lenuta Profire

Reviewer 3 Report

The manuscript in reference describes the synthesis, in-silico interaction with COX isozimes and in-vitro antioxidant capacities, PBS thermal denaturation effects and NO-releasing properties of a novel set of oxadiazole-2-thiol-containing indomethacin derivatives. The manuscript is interesting and has important results. However, some points should be addressed prior further consideration.

  1. A detailed scrutiny throughout the manuscript to revise/correct some grammar and stylistic issues is highly recommended.
  2. Lines 26-27: Authors should be aware that molecular docking is not a biological evaluation. It is only a very basic calculation of a ligand-protein interaction using scoring functions based on molecular mechanics and usually validated through robust statistics. Therefore, I recommend to give the proper scope of such an analysis to avoid confusing aim&scope of this part of the manuscript. Be consistent throughout the manuscript (e.g., lines 106-107, among others).
  3. Lines 27-29: Separate the in-silico study from the in-vitro studies. In addition, authors evaluated a thermal denaturation of serum protein effect instead an anti-inflammatory effect. So, I recommend to change this term here to provide the correct scope. Be careful on this and be consistent throughout the manuscript.
  4. Line 29: based on the previous comment, inhibitors cannot be stated or selected only by molecular docking simulations. I strongly recommend to revise and give the appropriated scope. Authors may be referring to a better interaction that putatively act as inhibitors exclusively if the inhibition mechanism is competitive and only in the case the docking site is the COX active site. Otherwise, molecular docking can only be referred as good/bad interaction to the target site through docking scores and no more. To select inhibitors from an in-house database, in-vitro enzymatic studies are the only way to do that. Be careful on this and be consistent throughout the manuscript.
  5. Lines 28, 38-40, 62-62, 72-73, 138-139, 243-244, 335-338, 339-341, among several others: Revise these sentences, since it are not totally understandable and/or have typos or grammar mistakes.
  6. Lines 83-92: I consider important this paragraph illustrating the importance of the oxadiazole moiety in bioactive molecules. However, I recommend to authors to add a good explanation and clear reason to use such a moiety in anti-inflammatory drugs, since such a reason/explanation is not clear and is very important to justify the proposed design.
  7. Lines 93-99: A scheme illustrating such four properties should be providing to add quality to the design.
  8. Scheme 1: Add the yield range for each conversion i-viii to give a productive span and scope of such transformations.
  9. Lines 129-137: Since this paragraph is part of the “Results and Discussion” section, some comments/discussion about yields (e.g., related to substituents effects) should be provided.
  10. Lines 159-162: RMSD variation for docking validation is very high using re-docking strategy. With this RMSD value (i.e., 2 A), the docking protocol is not adequately validated and not recommended. A detailed description about this validation should be provided. I also recommend to include a more robust validation protocol, for instance, through ROC curves to avoid decoys and select appropriately binders and non-binders.
  11. Lines 168-169: The docking result from autodock-based simulation is not strictly a Gibbs free energy (also line 1016, it is not a binding energy). It is more related to a docking score (as adequately mentioned in line 979) expressed in kcal/mol. Please, be careful on this and be consistent throughout the manuscript (e.g., Table 1 and other passages in manuscript).
  12. Lines 184-189 and Table 1: Be careful about the calculated selectivity index. This index is valid but I consider it is no adequately scoped. This index is more related to a better/worst docking score, based on favorable interactions depending on scoring functions, rather than an inhibitory effect, so its scope is very different to that explained in this paragraph. Be consistent throughout the manuscript.
  13. Figures 4, 5, 6, 7. The results of the post-hoc multiple comparison test (e.g., different letters indicating significant differences) should be included in this figure.
  14. Section 2.4 (and 3.4). Although the Mizushima's test is valid, I am not convinced that the term is properly used, since an anti-inflammatory effect is not observed. Such a test is more related to a relationship between the anti-inflammatory drugs and their interaction of serum proteins, but this relationship is not always followed due to some deviations, and authors do not know if test compounds are part of such deviations. Therefore, I recommend to authors change this subtitle by the adequate scope, such as “2.4. Thermal denaturation of serum protein”.
  15. Section “Results and Discussion”: This section has a descriptive analysis of the obtained results, but the discussion is poorly presented, since no comparison with other studies is performed. I recommend to authors to revise, perform and include a deeper discussion.
  16. Lines 391-393: More details about the operating conditions of HRMS should be provided, e.g., the ionization voltage, the ionization source and the analyzer.
  17. Line 423 and other compounds: I’m not sure if electron impact mas spectrometry (EI-MS) can provide an quasi-molecular ion [M+H]+, which is more typical for chemical or soft ionization (e.g., electrospray, ESI). So, I recommend authors to revise this and be consistent throughout the manuscript.
  18. Lines 1143-1145, Conclusions: I am not agree with this last sentence of “Conclusions” section, since it is not justified by results. No promising anti-inflammatory compounds were found, since no anti-inflammatory test was used. In addition, no promising compounds with reduction of gastrointestinal toxicity were found, since the observed effect is only related to a chemical test to assess the NO releasing, and no toxicity was evaluated neither NO releasing on gastrointestinal cells was explored. In my opinion, the promising compounds are adequately related to COX-2 in-silico interaction, DPPH radical-scavenging capacity, PBS denaturation-promoting effect and NO-releasing property. Therefore, I strongly recommend to authors provide the appropriately conclusions from an adequate scope of their obtained results to avoid overestimations and confusions.

Author Response

Dear Reviewer,

We are very grateful for your expertise and valuable evaluation of the quality of our manuscript and for your important and constructive suggestions. The corrections were highlighted in red in the manuscript using track changes. Please take in consideration our response to your comments/suggestions.

Thank you a lot for your expertise, your time and your availability!

Prof. dr. Lenuta Profire

Round 2

Reviewer 1 Report

The authors revised in details article entitled “Design, synthesis, in silico and in vitro studies for new nitric oxide-releasing indomethacin derivatives with 1,3,4-oxadioazole-2-tiol scaffold”. The authors characterized a number of organic molecules and led to advances in the production of indomethacin derivatives targeting the production of nitric oxide in a living cell. The 1,3,4-oxadiazole unit linked to a backbone containing a nitric oxide precursor appears to play the role of a bioisoster commonly used in medical chemistry and medicine. This very interesting article meets the requirements of articles addressed for Int. J. Mol. Sci. of MDPI and can be directed to the next stages of production after removing a few minor errors and ambiguities. Importantly, in all the cases presented in Scheme 3, the developed inhibitors bind the nitrate fragment to the active cavity of the tested enzymes, which suggests that a commonly used substrate trap strategy was applied with success. So, it seems that this part of the molecule served as a decoy, but this information is missing in the assumptions in the introduction. I present a few examples of similar strategies taken from the current scientific literature in order to briefly correct by authors in the introduction, to improve the quality of the publication: https://doi.org/10.1038/s41598-021-91809-9, https://doi:10.3390/molecules25051255, https://doi.org/10.15252/emmm.202012828, https://doi.org/10.1021/jm300687e, https://creativecommons.org/licenses/by-nc/4.0/.  

Below are a few questions (inaccuracies) for the authors and only minor typos that should be corrected before the next production stages:

In the line 298 of the first version (and in revision) there is: … positionis … , but should be at English: … positions … .

In the text revision of the line 410, the parentheses in the compound numbers 3p and 2p that follow the full compound name have been unnecessarily removed.

Comment: numbering after the full name of the molecules requires parentheses, and after the simplified name, the parentheses are redundant. In addition, in the proofreading, the brackets in the experimental part after the full name have been fully removed, which should be restored, only there is no need to enter brackets in bold style.

At line 551 of the first version is lack of the multiplicity and spin spin coupling constants at the 19F NMR spectroscopy of compound 4b, see line 430 of compound 3b.

At lines 673 and 676 of the first version (and in a revised firm) at compound 5b is JHF = 12.4 Hz, and JFH = 12.2 Hz, respectively, but at both cases should be J = 12.3 Hz. Please corrected. Comment: if hydrogen at 1H NMR coupled with fluorine, the fluorine at 19F NMR coupled with the hydrogen with exactly the same spin spin coupling constant. See for compound 8b.

At line 816 of the first version in 13C NMR at 130,2 ppm is triplet (t), but at the compound 8b is only one 19F atom (nuclear spin ½). Should be doublet (d).

For molecule 8d of the revised manuscript, the HRMS analysis has too large 0.01% an error. Please check, if you need to measure it again, if it fails, a new analytical sample is needed due to a large amount chlorine molecule, or alternatively classical elementary analysis also for Cl is needed.

In line 890 of the first version (and in a revised form) there is a multiplicity of dddd in 1H NMR for the signal at 6.88 ppm of two protons, but the given spin-spin coupling constant of 11.8 Hz suggest the appearance of two different signals different for 0.05 ppm apart for NMR apparatus with an operating frequency of 250 MHz. Please correct if necessary, especially since this spin-spin coupling constant suggests two diastereotopic protons (located on a common carbon atom).

Author Response

(The authors gave the same response as above.)

Reviewer 3 Report

Authors addressed adequately the previous comments. In addition, manuscript improved in quality, scope and content. I finally recommend to authors to revise a minor issue related Table 1 caption (i.e., place "Docking scores (kcal/mol)" instead of "The Gibbs free binding energies values (ΔG)" to avoid inaccuracy of docking results.

Author Response

(The authors gave the same response as above.)
